# A Survey of Deep Learning for Alzheimer's Disease

Qinghua Zhou [1], Jiaji Wang [1], Xiang Yu [1], Shuihua Wang [1] and Yudong Zhang [1,2,*]

1  School of Computing and Mathematical Sciences, University of Leicester, Leicester LE1 7RH, UK; qz105@le.ac.uk (Q.Z.); jw933@le.ac.uk (J.W.); xy144@le.ac.uk (X.Y.); shuihuawang@ieee.org (S.W.)
2  Department of Information Systems, Faculty of Computing and Information Technology, King Abdulaziz University, Jeddah 21589, Saudi Arabia
*  Correspondence: yudongzhang@ieee.org; Tel.: +44-754-870-0453

**Abstract:** Alzheimer's and related diseases are significant health issues of this era. The interdisciplinary use of deep learning in this field has shown great promise and gathered considerable interest. This paper surveys deep learning literature related to Alzheimer's disease, mild cognitive impairment, and related diseases from 2010 to early 2023. We identify the major types of unsupervised, supervised, and semi-supervised methods developed for various tasks in this field, including the most recent developments, such as the application of recurrent neural networks, graph-neural networks, and generative models. We also provide a summary of data sources, data processing, training protocols, and evaluation methods as a guide for future deep learning research into Alzheimer's disease. Although deep learning has shown promising performance across various studies and tasks, it is limited by interpretation and generalization challenges. The survey also provides a brief insight into these challenges and the possible pathways for future studies.

**Keywords:** deep learning; Alzheimer's disease; mild cognitive impairment; neural networks; recent advances





## 1. Introduction

Deep learning is a field of study that shows great promise for medical image analysis and knowledge discovery that is approaching clinicians' performance in a growing range of tasks [1–3]. The interdisciplinary study of Alzheimer's disease (AD) and deep learning have been a focus of interest for the past 13 years. This paper aims to survey the most current state of deep learning studies related to multiple aspects of research in Alzheimer's disease, ranging from current detection methods to pathways of generalization and interpretation. This section will first provide the current definition of AD and clinical diagnostic methods to provide the basis for deep learning. We then detail this interdisciplinary study's main areas of interest and current challenges.

### 1.1. Alzheimer's Disease and Mild Cognitive Impairment

Alzheimer's disease is the most common form of dementia and a significant health issue of this era [4]. Brookmeyer et al. [5] predicted that more than 1% of the world population would be affected by AD or related diseases by 2050, with a significant proportion of this cohort requiring a high level of care. AD usually starts from middle-to-old age as a chronic neurodegenerative disorder, but rare cases of early-onset AD can affect individuals of 45–64 years old [6]. AD leads to cognitive decline symptoms: memory impairment [7], language dysfunction [8], and decline in cognition and judgment [9]. An individual with symptoms may require moderate to constant assistance in day-to-day life, depending on the stage of disease progression. These symptoms severely affect patients' quality of life (QOL) and their families. Studies into cost-of-illness for dementia and AD reveal that the higher societal need for elderly care significantly increases overall social-economic pressure [10].

The biological process that leads to AD may begin more than 20 years before symptoms appear [11]. The current understanding of AD pathogenesis is based on amyloid peptide deposition and the accumulation and phosphorylation of tau proteins around neurons [12–14], which leads to neurodegeneration and eventual brain atrophy. Factors associated with AD include age, genetic predisposition [15], Down's syndrome [16], brain injuries [17], and cardiorespiratory fitness [18–20]. AD-related cognitive impairment can be broadly separated into three stages: (1) preclinical AD, where measurable changes in the brain, cerebral spinal fluid (CSF), and blood plasma can be detected; (2) mild cognitive impairment (MCI) due to AD, where biomarker evidence of AD-related brain change can be present; and (3) dementia due to AD, where changes in the brain are evident and noticeable memory, thinking and behavioural changes appear and impair an individual's daily function.

The condition most commonly associated with AD is mild cognitive impairment (MCI), the pre-dementia stage of cognitive impairment. However, not all cases of MCI develop into AD. Since no definite pathological description exists, MCI is currently perceived as the level of cognitive impairment above natural age-related cognitive decline [21,22]. Multiple studies have analyzed the demographics and progression of MCI and have found the following: 15–20% of people age 65 or older have MCI from a range of possible causes [23]; 15% of people age 65 or older with MCI developed dementia at two years follow-up [24]; and 32% developed AD and 38% developed dementia at five years follow-up [25,26]. The early diagnosis of MCI and its subtypes can lead to early intervention, which can profoundly impact patient longevity and QOL [27]. Therefore, better understanding the condition and developing effective and accurate diagnostic methods is of great public interest.

*1.2. Diagnostic Methods and Criteria*

The current standard diagnosis of AD and MCI is based on a combination of various methods. These methods include cognitive assessments such as the Mini-Mental State Examination [28–30], Clinical Dementia Rating [31,32], and Cambridge Cognitive Examination [33,34]. These exams usually take the form of a series of questions and are often performed with physical and neurological examinations. Medical and family history, including psychiatric history and history of cognitive and behavioral changes, are also considered in the diagnosis. Genetic sequencing for particular biomarkers, such as the APOE-e4 allele [35], is used to determine genetic predisposition.

Neuroimaging is commonly used to inspect various signs of brain changes and exclude other potential causes. Structural magnetic resonance and diffusion tensor imaging are widely applied to check for evidence of symptoms of brain atrophy. Various forms of computed tomography (CT) are also used in AD and MCI diagnosis. Regarding positron emission tomography (PET), FDG-PET [36] inspects brain glucose metabolism, while amyloid-PET is applied to measure beta-amyloid levels. Single-photon emission computed tomography [37] (SPECT) is likely to produce false-positive results and is inadequate in clinical use. However, SPECT variants can be potentially used in diagnosis, e.g., 99 mTc-HMPAO SPECT [38,39]. At the same time, FP-CIT SPECT can visualize discrepancies in the nigrostriatal dopaminergic neurons [40]. In neuroimaging, a combination of multiple modalities is commonly used to utilize the functionality of each modality.

New diagnostic factors of CSF and blood plasma biomarkers have been reported in the literature and have been deployed in clinical practice in recent years. There are three main CSF and blood plasma biomarkers: Amyloid-$\beta$42, t-tau, and p-tau. Other biomarkers include neurofilament light protein (NFL), and neuron-specific enolase (NSE, and HFABP [41,42]. CSF biomarkers are becoming a critical factor in AD diagnostic criteria in some practices. However, the actual 'ground truth' diagnosis of AD can only be made via post-mortem autopsy.

Before this century, the established diagnostic criteria were the NINCDS-ADRDA criteria [43,44]. These criteria were updated by the International Working Group (IWG) in 2007 to include requirements of at least one factor among MRI, PET, and CSF biomarkers [45]. A second revision was introduced in 2010 to include both pre-dementia and dementia phases [46]. This was followed by a third revision to include atypical prodromal Alzheimer's disease that shows cognition deficits other than memory impairment—IWG-2 [47]. Another independent set of criteria, the NIA-AA criteria, was introduced in 2011. These criteria include measures of brain amyloid, neuronal injury, and degeneration [48]. Individual criteria were introduced for each clinical stage, including pre-clinical [49,50], MCI [51,52], dementia [53,54], and post-mortem autopsy [55].

### 1.3. The Deep Learning Approach

Detailed preprocessing with refined extraction of biomarkers combined with statistical analysis is the accepted practice in current medical research. Risacher et al. [56] applied statistical analysis on biomarkers extracted using voxel-based morphometry and parcellation methods from T1-weighted MRI scans of AD, MCI, and HC. The study reveals statistical significance in multiple measures, including hippocampal volume and entorhinal cortex thickness. Qiu et al. [57] further confirmed this significance by analyzing regional volumetric changes through large deformation diffeomorphic metric mapping (LDDMM). Guevremont et al. [58] focused on robustly detecting microRNAs in plasma and used standardized analysis to identify microRNA biomarkers in different phases of Alzheimer's disease. This study and its statistical analysis yielded useful diagnostic markers reflecting the underlying disease pathology. The different biomarker information extracted was fed into statistical analysis methods with varying numbers of variables to detect changes in biomarkers in disease development [59]. Similar studies also employed other neuroimaging data, genetic data, and CSF biomarkers. These studies supported the use of MRI imaging biomarkers in AD [60] and MCI diagnosis [61], laying the basis for developing automatic diagnostic algorithms.

Machine learning has amassed great popularity among current automated diagnostic algorithms due to its adaptivity to data and the ability to generalize knowledge with lower requirements of expert experience. The study by Klöppel et al. [62] proved the validity of applying machine learning algorithms in diagnosing dementia through a performance comparison between the Support Vector Machine (SVM) classification of local grey matter volumes and human diagnosis by professional radiologists. Janousova et al. [63] proposed penalized regression with resampling to search for discriminative regions to aid Gaussian kernel SVM classification. The regions found by the study coincide with the previous morphological studies. These breakthroughs led to the development of many machine-learning algorithms for AD and MCI detection. Zhang et al. [64] proposed a kernel combination method for the fusion of heterogeneous biomarkers for classification with linear SVM. Liu et al. [65] proposed the Multifold Bayesian Kernelization (MBK) algorithm, where a Bayesian framework derives kernel weights and synthesis analysis provides the diagnostic probabilities of each biomarker. Zhang et al. [66] proposed the extraction of the eigenbrain using Welch's *t*-test (WTT) [67] combined with a polynomial kernel SVM [68] and particle swarm optimization (PSO) [69].

There has also been considerable interest in applying deep learning (DL), a branch of machine learning, to the field of AD and related diseases. Deep learning integrates the two-step feature extraction and classification process into neural networks, universal approximators based on backpropagation parameter training. [70]. Deep learning has made considerable advances in the domain of medical data, e.g., breast cancer [71], tuberculosis [72], and glioma [73]. Instead of hand-crafting features, models, and optimizers, deep learning leverages the layered structure of neural networks for the automated abstraction of various levels of features. For example, Feng et al. [74] used the proposed deep learning model to extract biomarkers for MRI in neuroimaging. The study demonstrates that the deep learning approach outperformed other neuroimaging biomarkers of amyloid and tau

pathology and neurodegeneration in prodromal AD. A visualization of the field of this survey is shown in Figure 1.

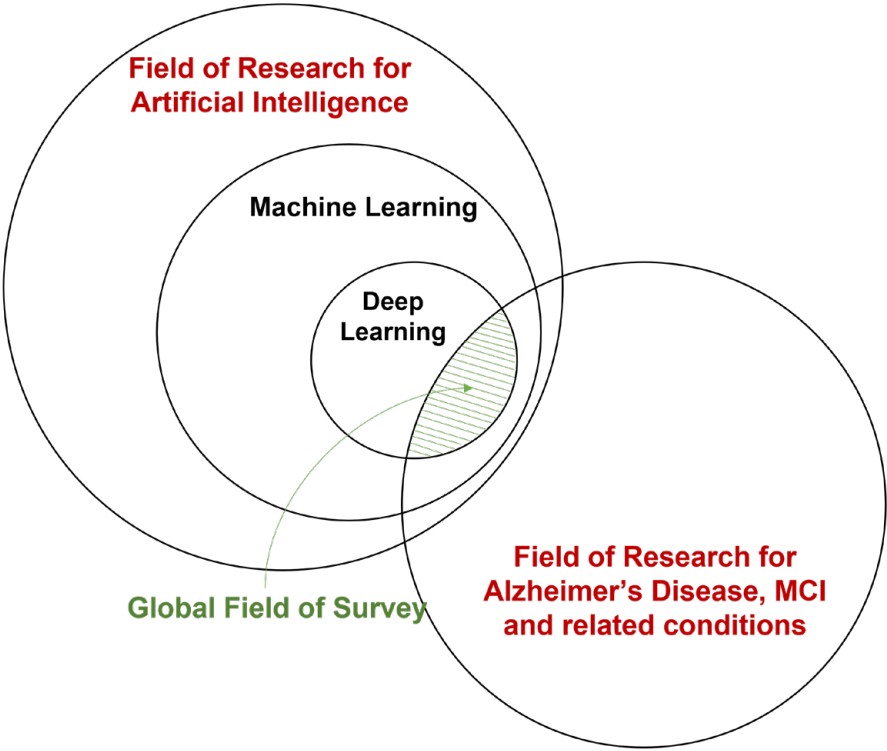

**Figure 1.** A broad overview of the field of this survey.

*1.4. Areas of Interest*

The primary aim of the surveyed deep learning studies in Alzheimer's and related diseases is detecting and predicting neurodegeneration to provide early detection and accurate prognosis to support treatment and intervention. The main interests of this interdisciplinary field can be roughly categorized into three areas:

1.  Classification of various stages of AD. This area targets diagnosis or efficient progression monitoring. Current studies mostly focus on AD, MCI subtypes, and normal cognitive controls (NC). A few studies contain the subjective cognitive decline (SCD) stage before MCI.
2.  Predicting MCI conversion. This area is mainly approached by formulating prediction as a classification problem, which usually involves defining MCI converters and non-converters based on a time threshold from the initial diagnosis. Some studies also aim at the prediction of time-to-conversion for MCI to AD.
3.  Prediction of clinical measures. This area aims at producing surrogate biomarkers to reduce cost or invasivity, e.g., neuroimaging to replace lumbar puncture. Prediction of clinical measures, e.g., ADAS-Cog13 [75] and ventricular volume [76], is also used for longitudinal studies and attempts to achieve a more comprehensive evaluation of disease progression and model performance benchmarking.

There are also other areas of interest, including knowledge discovery, where studies attempt to understand AD through data [77]. Another area of interest is phenotyping and sample enrichment for clinical trials of treatments [78], where DL models are used to select patients that will likely respond to treatment and prevent ineffective or unnecessary treatment [79]. Interest also lies in segmentation and preprocessing, where DL models are applied to achieve higher performance or efficiency than conventional pipelines [80].

*1.5. Challenges in Research*

There is uncertainty in the diagnosis or prognosis of AD or related diseases with still developing diagnostic criteria and scientific understanding. DL-based approaches have already shown potential in the above areas of interest; however, there exists room for improvement and a range of challenges:

1. Numerical representation of the differences between AD stages. Monfared et al. [81] calculated the range of Alzheimer's disease composite scores to assess the severity of the cognitive decline in patients. Sheng et al. [82] made multiple classifications and concluded that the gap between late mild cognitive impairment and early mild cognitive impairment was small, whereas a greater difference exists between early and late MCI patients. Studies comparing clinical and post-mortem diagnoses have shown 10–20% false cases [83]. In addition, autopsy studies in individuals who were cognitively normal for their age found that ~30% had Alzheimer's-related brain changes in the form of plaque and tangles [84,85]. Sometimes the signs that distinguish AD, for example, brain shrinkage [86], can be found in a normal healthy brain of older people.

2. Difficulty in preprocessing. Preprocessing medical data, especially neuroimaging data, often requires complex pipelines. There is no set standard for preprocessing, while a broad range of processing options and relevant parameters exist. Preprocessing quality is also vastly based on the subjective judgment of clinicians.

3. Unavailability of a comprehensive dataset. Though the amount and variety of data available for AD and related diseases are abundant compared with many other conditions, the number of subjects is only moderate compared with large datasets such as Image-Net and is below the optimal requirements for generalization.

4. Differences in diagnostic criteria. The diagnostic criteria, or criteria for ground truth labels, can differ significantly between studies, especially in prior studies before new methods of diagnosis (e.g., CSF biomarkers [87] and genetic sequencing [88]) became accessible.

5. Lack of reproducibility. Most frameworks and models are not publicly available. Without open-source code, implementation details such as specific data cohort selection, preprocessing procedures and parameters, evaluation procedures, and metrics are usually lacking. These are all factors that can significantly impact results. Additionally, few comprehensive frameworks are designed for benchmarking different models based on the same preprocessing/processing and testing standards [89,90].

6. Lack of expert knowledge. Researchers adept at using DL often have no medical background, while medical data are significantly more complicated than natural images or language data. Therefore, these researchers lack expert knowledge, especially in preprocessing and identifying brain regions of interest (ROIs).

7. Generalizability and interpretability. Current DL models are plagued by information leakage and only provide limited measures of generalizability, the model's performance in real-world populations. The inherent 'black box' nature of neural networks impedes the interpretation of model functions and the subsequent feedback of knowledge for clinicians [91].

8. Other practical challenges include the subjectivity of cognitive assessments, the invasiveness of diagnostic techniques such as a lumbar puncture to measure CSF biomarkers and the high cost of neuroimaging such as MRI.

By analyzing the frequency of occurrence, influencing factors, and potential impact on research results for each challenge based on evidence and observations in the literature, we assign weights to each challenge in Table 1.

**Table 1.** Summary of challenges in applying DL to AD.

| Challenge | Description | Weight (1–5) |
|---|---|---|
| Numerical representation of AD stages | Variability in Alzheimer's disease composite scores and difficulty distinguishing between stages of cognitive impairment. | 3 |
| Difficulty in preprocessing | Complex pipelines for preprocessing medical data, lack of standardization, subjective judgment of clinicians. | 3 |
| Unavailability of a comprehensive dataset | Abundance of data for AD but moderate number of subjects, below optimal requirements for generalization. | 2 |
| Difference in diagnostic criteria | Variations in diagnostic criteria and ground truth labels between studies, impacting comparability of results. | 3 |
| Lack of reproducibility | Lack of publicly available frameworks, implementation details, and comprehensive benchmarking standards. | 4 |
| Lack of expert knowledge | Researchers with DL expertise may lack medical background, particularly in preprocessing and identifying brain regions. | 2 |
| Generalizability and interpretability | Limited measures of generalizability, 'black box' nature of neural networks, hindering model interpretation and feedback. | 5 |
| Practical challenges | Subjectivity of cognitive assessments, invasiveness and cost of diagnostic techniques such as lumbar puncture and MRI. | 3 |

*1.6. Survey Protocol*

This survey covers DL studies related to AD or related diseases from 2010 to 2023. To identify literature related to our focus, we first queried the online libraries of IEEE, New York, NY, USA, Springer, Berlin/Heidelberg, Germany, and ScienceDirect, Amsterdam, The Netherlands and then concentrated search on:

1. Recognized journals, including Brain, Neuroimage, Medical Image Analysis, Alzheimer's and Dementia, Nature Communications, and Radiology.
2. Conferences in computer vision and deep learning, including ACM, NeurIPS, CVPR, MICCAI, and ICCV.

The full list of search keywords is as follows: "Alzheimer's", "AD", "Dementia", "Mild Cognitive Impairment", "MCI", "Neural Networks", "Deep Learning", "Machine Learning", "Learning", "Big Data", "Autoencoders", "Generative", "Multi-Modal", "Interpretable", "Explainable". These keywords were used independently or in combination during the search process, which yielded 360 papers from various sources. A two-stage selection was performed, where the following conditions were first used to select the papers:

1. Related to Alzheimer's disease, MCI, or other related diseases.
2. Related to deep learning, with the use of neural networks.
3. Contains valid classification/prediction metrics.
4. Utilizes a reasonable form of validation.
5. Written in English or contains a valid translation.
6. Contains a minimum of 180 individual subjects.

An additional constraint of subject number was applied in this survey, where 180 subjects correspond to a 0.01 chance of having an approximately 10% fluctuation in accuracy in generalization according to derivations from Hoeffding inequality [92]. However, this is only a basic requirement since the approximate generalization bound depends on the data available for evaluation and the independence assumptions between the classifier parameters and data. This condition is relaxed for studies using uncommon data types and functional MRI, where the available data are often limited compared with standard data types, e.g., MRI and PET. This selection stage yielded a total of 165 papers.

After the first stage of selection, the papers were evaluated on the year of publication, data source and type, preprocessing technique, feature extraction techniques, model architecture, platform, optimization protocol, and evaluation details. Papers from unknown sources or studies with apparent errors, e.g., information leakage, were excluded from the selection. This selection stage yielded a total of 83 papers. Similar to previous surveys, this survey mainly focused on neuroimaging data [93–96]. This review also expands on the work by Wen, Thibeau-Sutre, Diaz-Melo, Samper-González, Routier, Bottani, Dormont, Durrleman, Burgos and Colliot [89], which focuses on convolutional neural networks, to a broader range of supervised and unsupervised neural networks, including recent advances in graph and geometric neural networks. The survey protocol is visualized in Figure 2.

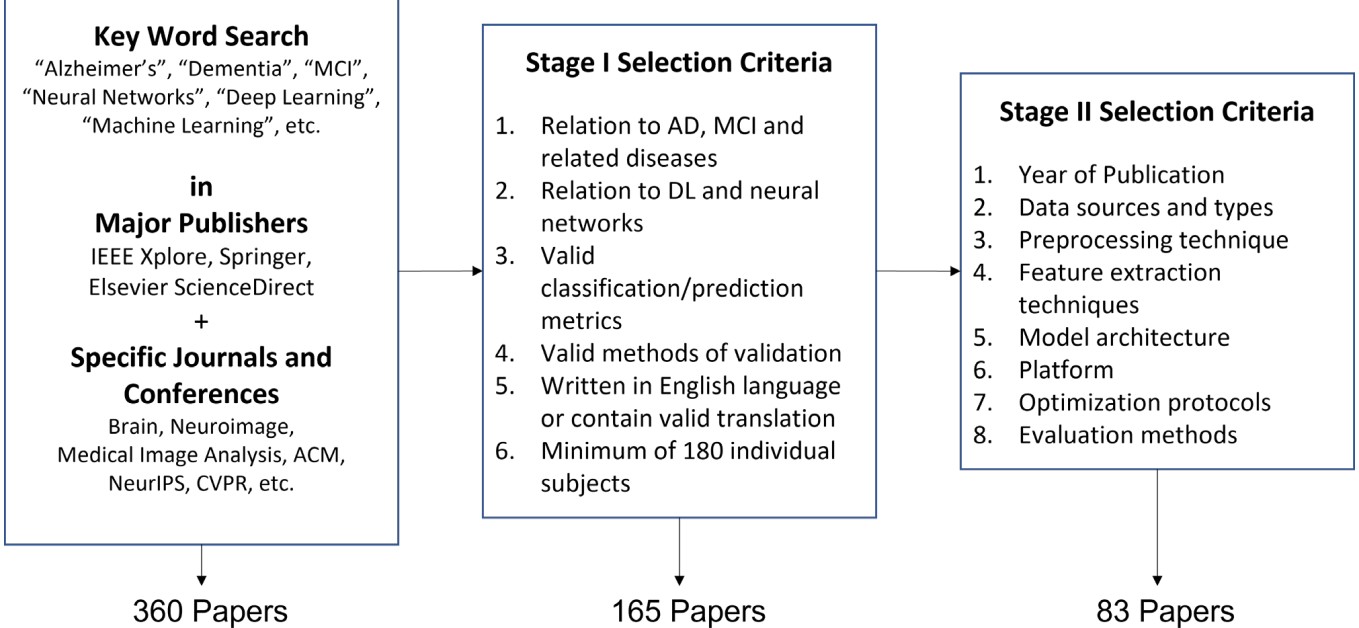

**Figure 2.** Survey Protocol.

The paper is organized as follows: Section 2 introduces the data types implemented in deep learning research and potential data sources; Section 3 provides detailed summaries of data preprocessing methods for the main data types, followed by four categories of data processing for neural network data input in Section 4. Sections 5–7 constitute the main body of deep learning architectures and methods included in this review, categorized into unsupervised, semi-supervised, and supervised learning methods; typical models and recent advances are included in each category, including recent developments in generative models, recurrent and graph neural networks. Section 8 introduces various techniques, including transfer learning, ensemble learning, and multi-modal fusion. Section 9 details training and evaluation protocols, while Sections 10 and 11 lay possible pathways for future research in interpretability and generalization. A taxonomy of the survey is shown in Figure 3.

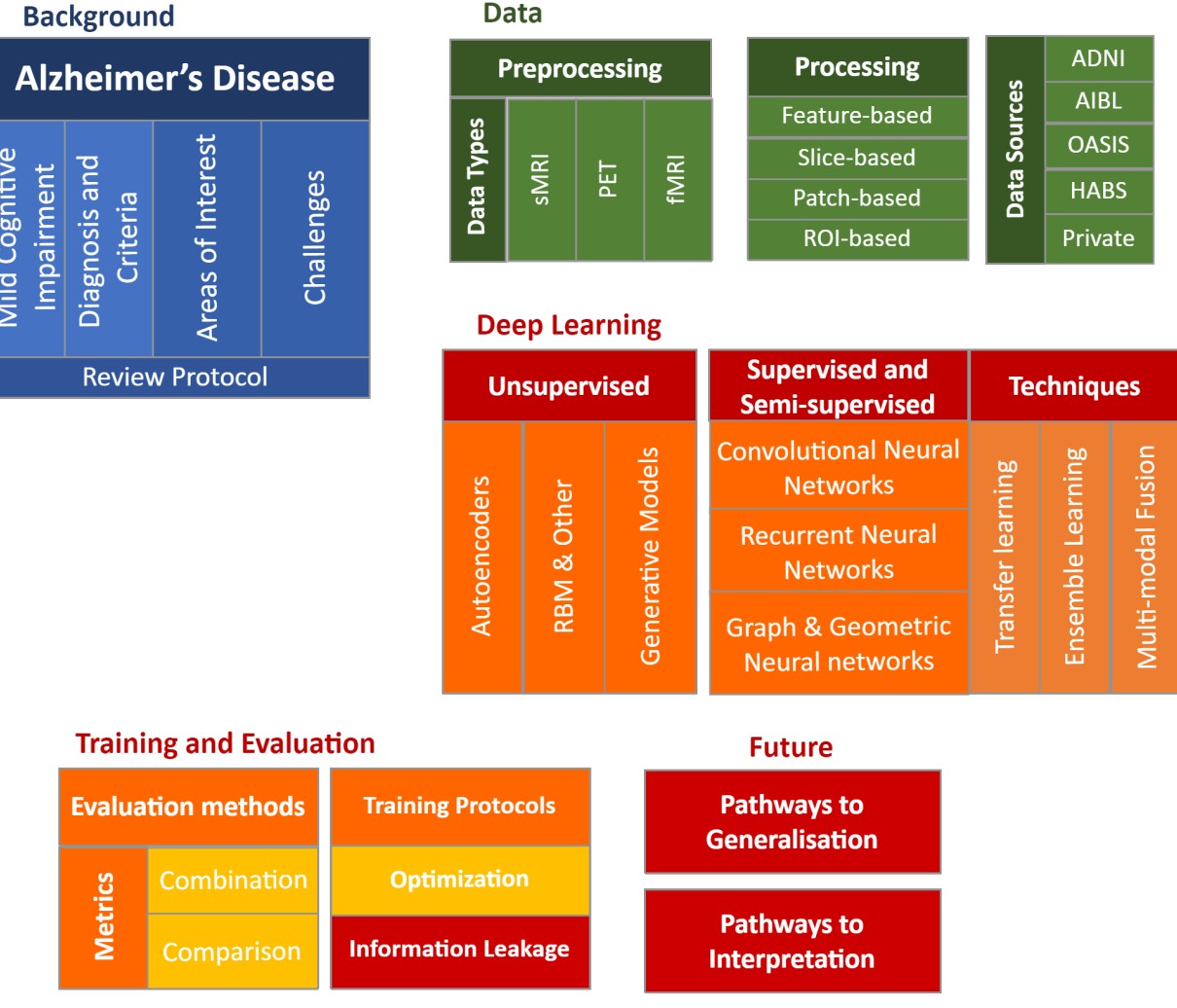

**Figure 3.** Overview of the survey content.

## 2. Data Types and Sources

Data issues are a core aspect of the deep learning approach, and the type and quantity of data directly impact model performance and potential generalizability. A large variety of data from numerous sources have been utilized in the reviewed studies. In this section, we summarize the main types of data available and the sources of these data.

### 2.1. Types of Data

The available data can be categorized into longitudinal data and cross-sectional data. Longitudinal data correspond to a subject's disease progression data collected over time, while cross-sectional data are single-instance data that are time-independent. Longitudinal data can also be treated as independent data, time-series data, or comparative data. Demographics (Demo) is often a form of meta-data collected alongside other exams, primarily information regarding age, gender, and education. Neuroimaging data of various modalities are commonly collected. Common modalities include PET, MRI, and CT for diagnosis, and 3D-MRI [97], fMRI [98], and SPECT for research purposes. Various forms of cognitive assessments (CA) are also commonly available, including MMSE [29], CDR [99], ADAS-Cog [100,101], logical memory test [102,103], and postural kinematic analysis [104,105]. CSF, blood plasma biomarkers [106,107], and genetic data are available from several sources. Other less common data types include electroencephalography (EEG) for brain activity monitoring [106,107], mass spectra data collected through surface-enhanced laser desorption and ionization assay of saliva [108], and retinal imaging for

abnormalities [109]. Electronic health records have also been studied to screen dementia and AD [110,111]. Alternative data types such as speech [112,113], activity pattern monitoring [114,115], and eye-tracking [116] have also been studied with a deep learning approach. Few deep-learning-related comparative studies have been performed between different data modalities and types, especially for the less common data types [117].

*2.2. Sources of Data*

Several open libraries have been created in the past two decades, providing easier access for researchers to available data on subjects with AD or related diseases. One of the main libraries is Alzheimer's Disease Neuroimaging Initiative (ADNI) [118], a large longitudinal study aiming to develop novel biomarkers to detect AD and monitor disease progression. The original ADNI cohort was collected from 2004 to 2010 and contains T1-weighted MRI [119], FDG-PET, blood, and CSF biomarkers from 800 subjects [120]. Additional cohorts, ADNI-Go and ADNI-2, extended the longitudinal study of ADNI-1 while also encompassing a broader range of the stages of AD, adding 200 new subjects with early MCI [121]. A fourth cohort, ADNI3, with additional modalities targeting tau protein tangles, started in 2016 and is due to complete in 2022 [122].

ADNI is the most commonly used open library available for neuroimaging data. Another commonly used open library is the Open Access Series of Imaging Studies (OASIS), which includes a cross-sectional cohort (OASIS-1), a longitudinal cohort (OASIS-2) of demented or non-demented subjects MRI, and an additional longitudinal cohort (OASIS-3) provides MRI and PET in various modalities of 1098 subjects of normal cognition or AD [123]. While ADNI contains genomic data, OASIS only contains neuroimaging and neuropsychology data, i.e., cognitive assessments.

Other open libraries include the Harvard Aging Brain Study (HABS) [124] and Minimal Interval Resonance Imaging in Alzheimer's Disease (MIRIAD) [125]. These open libraries are essential in propagating studies in machine learning or deep learning for AD research. There also exist several local studies modeled similar to ADNI for data compatibility, including Japan ADNI (J-ADNI) [126], the Hong Kong Alzheimer's Disease Study [127], and the Australian Imaging Biomarkers and Lifestyle Study of Ageing (AIBL) [128]. Various institutes have established platforms to provide information and efficient access to available databases and libraries, including NeuGRID [129,130] and the Global Alzheimer's Association Interactive Network [131]: http://www.gaain.org/ (accessed on 17 April 2023). We provide a shortlist of selected data sources in Table 2.

An alternative source of data for ML practitioners and researchers alike are the challenges hosted either by ADNI or other institutions, such as CADDementia [132], TADPOLE [133,134], DREAM [135], and the Kaggle international challenge for automated prediction of MCI from MRI data. These challenges may provide pre-selection or preprocessed data, reducing the need for expert knowledge. A few studies have proposed to use brain age as a surrogate measure of cognitive decline and utilized databases of cognitively normal individuals, including UKBioBank [136], NKI, IXI [137], LifespanCN [138], and the Cambridge dataset [139]. Other sources of data that may be available include data from the International Genomics of Alzheimer's Project (IGAP) [140], the Korean Longitudinal Study on Cognitive Aging and Dementia (KLOSCAD) [141,142], the INSIGHT-preAD study [143,144], the Imaging Dementia—Evidence for Amyloid Scanning (IDEA) study [145], and the European version of ADNI—AddNeuroMed [146,147]. Institutes that hold private data collections of AD or related diseases include the National Alzheimer's Coordination Center, the Biobank of Beaumont Reference Laboratory, and IRCCS. As one of the most significant health crises of the era, many studies have collected data for AD and related diseases; therefore, the above-listed sources include only those most commonly used in reviewed literature and examples of alternative sources.

**Table 2.** Sources of AD and dementia data.

| Library | Number of Subjects | Modalities | Link |
|---|---|---|---|
| ADNI | 2750 | MRI, PET, CSF, Genetic | http://adni.loni.usc.edu/ (accessed on 17 April 2023) |
| OASIS | 1300+ | MRI, PET | https://oasis-brains.org/ (accessed on 17 April 2023) |
| AIBL | 1100+ | MRI, PET, CSF, Genetic | https://aibl.csiro.au/ (accessed on 17 April 2023) |
| NACC | 47,000+ | Neuropathology, Genetic | https://www.alz.washington.edu/ (accessed on 17 April 2023) |
| EDSD | 471 | MRI, DTI, Genetic | https://www.neugrid2.eu/ (accessed on 17 April 2023) |
| ARWIBO | 2700+ | MRI, PET, Genetic | http://www.arwibo.it/ (accessed on 17 April 2023) |
| HABS | 290 | MRI, PET, Genetic | https://habs.mgh.harvard.edu/ (accessed on 17 April 2023) |
| KLOSCAD | 6818 | MRI, QOL, Behavioral | http://kloscad.com/ (accessed on 17 April 2023) |
| VITA | 606 | MRT, Genetic | https://www.neugrid2.eu/ (accessed on 17 April 2023) |

## 3. Data Preprocessing

The deep learning approach can replace the feature-crafting step of machine learning and reduce the need for preprocessing. Data types such as clock-drawing test images [148], activity monitoring data [115], and speech audio files [112] can be processed in a similar way to natural images and time-series data. However, for the prevalent neuroimaging data, due to the complexity of data and the variety of established pipelines, data preprocessing is a significant component in current DL studies. This survey will focus on imaging data, the most prevalent data category in the intersection between AD and deep learning. Differences in the organization of data adds to the difficulty of preprocessing. Gorgolewski et al. [149] proposed the Brain Imaging Data Structure (BIDS) repository structure. Conversion to a standard data structure such as BIDs is essential when using multiple modalities and data sources.

### 3.1. Structural MRI Data

MRI is a safe, non-invasive medical imaging technique. High-quality medical images can be generated with good spatial resolution while minimizing patient harm using a powerful magnetic field, radio waves, and a computer. Structural MRI (sMRI) and functional MRI (fMRI) are different MRI techniques used to study the brain. sMRI is a non-invasive brain imaging technique that can investigate changes in brain structure [150]. Changes in brain structure due to worsening cognitive impairment may include atrophy of specific brain regions, loss of brain tissue, and changes in the shape and size of certain brain structures [151,152].

MRI machines are highly complex medical equipment that can vary individually. Inhomogeneity of the B1-field in MRI machines can cause artifact signals known as the bias field. Bias field correction is often the first step in MRI data preprocessing [153,154], usually using B1-scans to correct for the non-uniformity in the MR image. Similarly, gradient non-linearity can be corrected with displacement information and phase mapping, e.g., Gradwarp. These corrections are often in-built into the MRI systems, and its outputs are often the raw data available from data sources. Intensity normalization is essential to mitigate the difference between multiple MRI machines, especially in large-scale multi-center studies or when combining data from multiple sources. The most common method

found in AD-related papers is the N3 nonparametric non-uniform intensity normalization algorithm [155,156], a histogram peak sharpening algorithm that corrects intensity non-uniformity without establishing a tissue model. In some studies, motion correction is used to correct for subject motion artifacts produced during scanning sessions [157].

Brain extraction is a common MRI preprocessing component. It is the removal of non-brain components from the MRI scan. Skull-stripping removes the skull component, e.g., through bootstrapping histogram-based threshold estimations. Other similar procedures include cerebellum removal and neck removal. The extracted brain images are often registered to a brain anatomical template for spatial normalization, usually performed after brain extraction. Registration can be categorized based on the deformation allowed into affine registration and non-linear registration. Affine registration includes linear registration, while non-linear registration allows for local deformations [158–160]. A standard template used is MNI-152 based on 152 subjects [161], while some studies use alternative templates such as Colin27. A potential challenge in this process is that the selected control subjects' age does not match the AD subjects' older age and corresponding brain atrophy. Some studies resolve this issue by constructing study-specific template space based on training data, which can also be aligned with standard templates. Other alignments include AC-PC correction, the alignment of the images with the anterior commissure (AC) and posterior commissure (PC) on the same geometric plane. AC-PC correction can be performed with resampling to $256 \times 256 \times 256$ and intensity normalization using the N3 algorithm with MIPAV. Studies have shown that linear or affine normalization is potentially sufficient for deep learning models [162,163], while other studies have shown that non-rigid registration can improve performance.

Another potential MRI preprocessing procedure is brain region segmentation, the division of the brain MRI into known anatomical regions. This step is usually performed to isolate brain regions related to AD, e.g., grey matter of the medial temporal lobe and the hippocampal region. Segmentation can be performed manually by outlining bounding boxes or precise pixel boundaries. Ideal practices include randomizing the samples and segmenting multiple times or segmentation with multiple expert radiologists [164]. However, manual segmentation is time intensive and not suitable for large datasets. Automated algorithms such as FSL FIRST [165] and the FreeSurfer pipeline can perform segmentation by registering to brain atlases, e.g., AAL. Other methods include using RAVENS maps produced by tissue-preserving image wrapping methods [166] and specific region segmentation, e.g., hippocampus segmentation with MALPEM [167]. With segmentation-based neural networks, multiple studies have applied the deep learning approach to hippocampus segmentation [168,169].

In AD-related studies, downsampling is often performed after preprocessing to reduce the dimensionality of input into the neural network, directly affecting the number of parameters and computational cost and achieving uniformity in input dimensions [170]. Smoothing is also often performed to further improve the signal-to-noise ratio [166] but it results in lower amplitude and increases peak bandwidth. Age correction also considers normal brain atrophy due to increasing age, similar to atrophy due to AD. A potential method to correct this effect is via a voxel-wise linear regression model after registration, which benefits overall model performance [167].

### 3.2. PET Data

PET utilizes a radioactive tracer to study the activity of cells and tissues in the body [171]. When studying neurological disorders, the tracer binds to specific proteins associated with the disease, such as amyloid beta, a hallmark of AD [172], and tau in the case of AD [173,174]. It can also help identify changes in glucose metabolism, which is altered in the brains of Alzheimer's patients. The preprocessing of PET images is similar to the preprocessing of structural MRI images described in Section 3.1. In AD-related studies, PET data are often used with MRI data due to combined collection in major studies, e.g., ADNI. Preprocessing up to image registration and segmentation is first performed

on the MRI image, while the PET images are registered to the corresponding MRI images through rigid alignment [166]. The post-segmentation steps of downsampling and smoothing are similar to those performed on MRI images. Studies independent of MRI follow either simplified preprocessing methods similar to MRI preprocessing [80,175,176] or only minimal preprocessing [177].

### 3.3. Functional MRI Data

Functional MRI (fMRI) is a type of magnetic resonance imaging designed to measure brain activity by monitoring blood flow within the brain. Instead of static, single-instance structural MRI, fMRI is temporal, consisting of a series of images. fMRI is used to study changes in brain function related to the disease. These changes can encompass altered connectivity between distinct regions of the brain [178] and variations in how the brain reacts to stimuli [179]. The fMRI can investigate alterations in memory and attention associated with cognitive impairment in MCI and AD [180]. Both sMRI and fMRI can be utilized to monitor the progression of the disease by detecting changes in specific brain regions over time [181,182].

Therefore, preprocessing steps in addition to the preprocessing procedures for structural MRI mentioned in Section 3.1 are required. Slice time correction is required to achieve the time-series exact timing, where fMRI may need to be first corrected for the temporal offset between each scan instance. More extended periods of fMRI scanning and the collection of multiple images in a single session increase the chance of head motion artifacts. Therefore, fMRI scans require additional filtering or correction for motion. Head motion correction of fMRI is usually performed through the spatial alignment to the first scan, or scan of choice, before spatial normalization. High-pass and low-pass filters can also be introduced to the temporal domain to control the fMRI data frequency and period [183]. The preprocessing of fMRI data can be automated using the SPM REST Toolkit, DPABI, or FreeSurfer. Data redundancy reduction methods are often applied to fMRI data; these can be categorized as methods based on common spatial pattern (CSP) or brain functional network (BFN). CSF-based methods produce spatial filters that maximize one group's variance while minimizing another [184]. BFN-based methods use ROI segmentation to construct a brain network where the ROI features are vertices and the functional connections are edges. Brain networks can also be constructed by calculating ROI correlations after segmentation [185]. A recent study has also applied the deep learning approach to construct weighted correlation kernels integrated into neural network architecture to extract dynamic functional connectivity networks [186].

### 4. Data Processing

Data processing is essential to the deep learning approach, significantly influencing model architecture and performance. Compared with traditional machine learning feature extraction, data processing for deep learning focuses on processing input data to neural networks instead of establishing quantified representations. Data processing aims to preserve and emphasize critical discriminatory information within the preprocessed or raw data while standardizing the input for model readability across samples and modalities. The processing can be categorized into common types of model inputs. A basic summary of the most commonly used input types is illustrated in Figure 4.

### 4.1. Feature-Based

Feature-based approaches are performed on individual features of the provided data. For neuroimaging, this is also known as the voxel-based approach [96], which is applied to individual image voxels of spatially normalized images. Space co-alignment between images is also essential to ensure comparability between individual voxels across the dataset. To limit the amount of input information, tissue segmentation of the grey matter probability maps is often performed. Machine learning extraction of texture, shape, or other features can also be performed to reduce dimensionality and form an ML-DL hybrid approach [187].

Voxel-based methods for neuroimaging data retain global 2D or 3D information but ignore local information as it treats the entire brain uniformly, regardless of anatomical features. For 3D scans and genetic data with large transcription quantity, higher dimensions of input result in high computational cost; dimensionality reduction through either feature selection or transformations is common. Feature-based approaches are used for most alternative data types such as cognitive assessments, CSF, serum, and genetic biomarkers. Longitudinal data for these types of time-series data such as EEG, activity, and speech requires more stringent processing for sample completeness, e.g., imputation for missing data and time-stamp alignment [188].

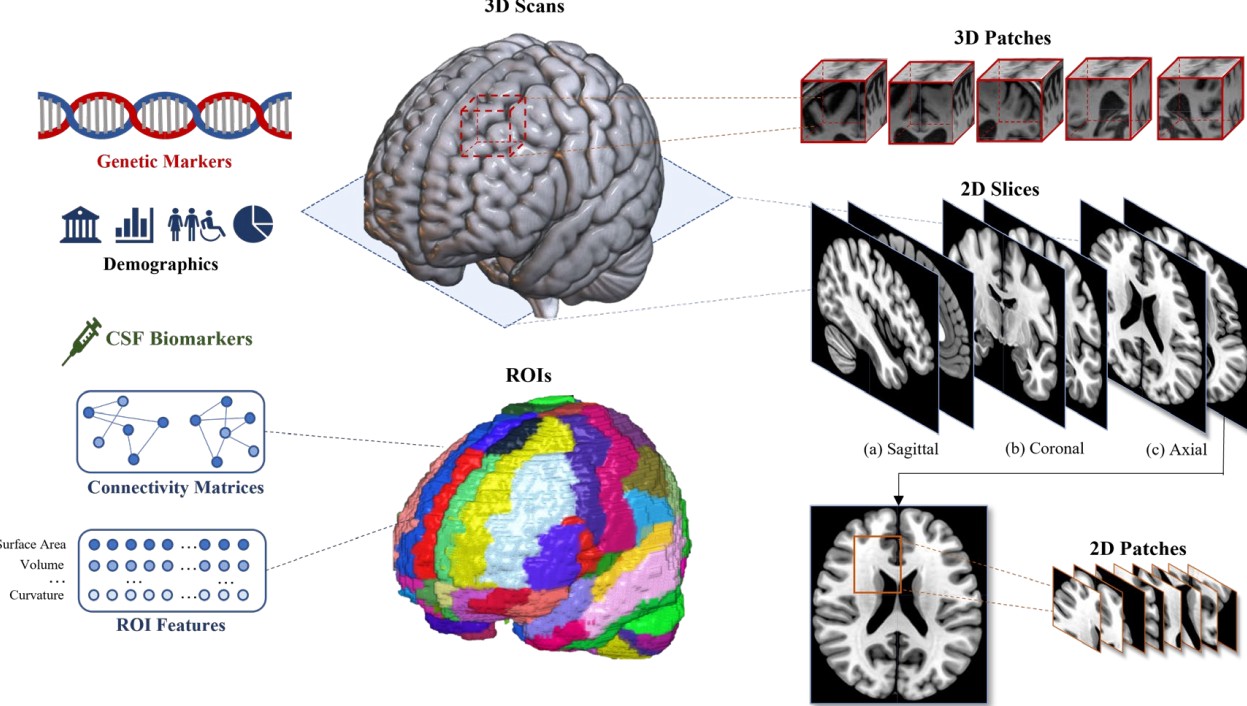

**Figure 4.** Summary of the most commonly used data types, where an sMRI is provided as an example of neuroimaging data. Feature-based data: demographics, CSF biomarkers, genetic markers, and 3D scans. Slice-based data: 2D slices from three views. Patch-based data: 2D and 3D patches. ROI-based data: ROI features and connectivity matrices.

### 4.2. Slice-Based

Slice-based approaches use 2D images or data. For 3D information, slice-based approaches assume that 2D information is a sufficient representation of the required information. Practical clinical diagnosis is often based upon a limited number of 2D slices instead of a complete 3D image. Some studies extract single or multiple 2D slices along the sagittal, axial, and coronal planes from the 3D scan. Slices from the axial plane are most commonly extracted, whereas the coronal view might contain the most critical AD-related regions. The selection of slices from 3D scans usually focuses on a particular dissection of the brain and the anatomical components it contains, e.g., the sagittal slices of the hippocampus are a known region of interest. Some studies used sorting procedures to find the most valuable slices, e.g., entropy sorting with greyscale histograms (Choi and Lee 2020). Slice-based approaches can be less computationally expensive than feature-based approaches with limited information quantity. However, the drawback of slice-based approaches is the loss of global and 3D geometric structures. Studies attempt to compensate for this loss by using multiple slices from multiple views, e.g., slices from three projections that show the hippocampal region [189], and multiple modalities, e.g., combining slices from MRI and PET.

### 4.3. Patch-Based

Instead of using all features or 2D slices, we can use regions of predefined size as input to the model, which is known as the patch-based approach. These regions can be 2D or 3D to suit model requirements [166]. Lin, Tong, Gao, Guo, Du, Yang, Guo, Xiao, Du and Qu [167] combined 2D greyscale patches of the hippocampal region into RGB patches. The patch-based approach can provide a larger sample size, equivalent to the number of patches, in the training procedure. Individual patches have a smaller memory footprint with lower input dimensions, reducing the computational resources required for training. However, additional resources for reconstructing sample-level results will mean costs to efficiency during testing and application. The challenge of patch-based approaches is capturing the most informative regions. Region selection is a vital component in this category; this includes the size of patches, the choice of overlap between the patches, and the choice of essential patches. Studies have attempted to use voxels' statistical significance [56,190] to find patching regions, while landmark-based methods perform patching around anatomically significant landmarks [191]. These patch-based approach is an intermediate form of voxel-based and ROI-based methods. Various kinds of approaches require patch-level data, including the use of a single or a low number of patches from each image for low input dimension models used to localize atrophy [192], patch-level sub-networks for hierarchical models [166], and ensemble learning through networks trained on defined regions.

### 4.4. ROI-Based

Patch-based methods have predefined regions, and sizes of extraction are often rigid, while ROI-based methods focus on anatomical regions of interest within the brain. These ROIs of anatomical function are often finely selected in the preprocessing stage of registration to brain atlases. The most common atlas used among the reviewed studies is the Automated Anatomical Labeling (AAL) atlas, which contains 93 ROIs. Other atlases include the Kabani reference work [193] and the Harvard-Oxford cortical and subcortical structural atlases [194,195]. Elastic registration, such as HAMMER, has higher registration performance [196,197]. After ROI extraction, the reviewed studies commonly use GM tissue mean intensities, or volumes, of brain ROIs as features from PET, MRI, fMRI or other modalities [198]. Other measures include subcortical volumes [199,200], grey matter densities [201,202], cortical thickness [203,204], brain glucose metabolism [205,206], cerebral amyloid-β accumulation [207,208], and the average regional CMRGlc [209] for PET. The hippocampus is of particular interest in the reviewed papers; ROI-based methods have used 3D data and morphological measurements of its cortical thickness, curvature, surface area, and volume. Aderghal, Benois-Pineau and Afdel [189] proposed using both left and right hippocampal regions through flipping regions along the median plane. The relationships between ROIs are also used as standard input; the correlation between regions provides connectivity matrices that are often divided into cortical and subcortical regions [210].

ROI-based methods are closely linked to anatomical regions and have high interpretability and clinical implementability. However, the close link to a priori knowledge limits its potential in explorative studies. The computational cost is usually between that of the voxel and slice-based approaches, but ROI-based methods can maintain local 3D geometric information. Hierarchical neural network frameworks containing sub-networks at each representation level have also been proposed with effective network pruning to retain complete information [192].

*4.5. Voxel-Based*

Voxel-based approaches are feature-based approaches that focus on the analysis of individual voxels, which are the three-dimensional pixels that make up a medical image [211]. These voxels represent discrete locations in the brain [212], and their size and number can be adjusted to balance computational efficiency and spatial resolution [213]. Compared with slice-based approaches, voxel-based methods can capture the three-dimensional structure of the brain and its changes, which may not be evident in two-dimensional slices. Due to the complexity of brain structure and differences between subjects, spatial co-alignment (registration) is essential [214]. Registration is the process of spatially aligning image scans to an anatomical reference space [215]. This process involves aligning MRI images of different patients or the same patient at different time points to a standardized template representing a common anatomical space [216]. Many studies segment the aligned images into different tissue types, such as gray matter, white matter, and cerebrospinal fluid, using unique signal features of different tissue types before applying the model [217,218]. Comparing gray and white matter across groups or time points can be a sensitive method for detecting subtle changes in brain structure. However, voxel-based approaches also have limitations. One major limitation is the requirement for high spatial resolution. The paper [219] utilizes functional network topologies to depict neurodegeneration in a low-dimensional form. Furthermore, functional network topologies can be expressed using a low-dimensional manifold, and brain state configurations can be represented in a relatively low-dimensional space.

## 5. Introduction to Deep Learning

Deep learning (DL) is a branch of machine learning which implements universal approximators of neural networks [70,220], a modern development of the original perceptron [221,222] with chain rule-derived gradient computation [223,224] and backpropagation [225,226]. The fundamental formulation of the neural network can be represented through the formulation of a classifier:

$$y = f'(x) \tag{1}$$

where $x$ is the data. The function $f'$ represents the ideal mapping between the input and the underlying solution $y$. A neural network defines a mapping $f(x, \theta)$ that provides an approximation of $f'$ by adjusting its parameters $\theta$. This adjustment can be considered a form of learning. For learning, a loss function, $L(f(x; \theta), y)$, can be constructed through the relation between the ideal output and the current output of the neural network. Backpropagation through derivatives of the loss function provides a means of updating the parameters for the learning process with a learning rate of $\epsilon$. DL can abstract latent feature representations with minimal manual interference. Features generated by DL cover the hierarchy of low- to high-level features that extend from lines, dots, or edges to objects or characteristic shapes [227].

$$\theta \leftarrow \theta - \epsilon \frac{\partial L(x, \theta)}{\partial \theta}, \tag{2}$$

Advances in deep learning have achieved performance comparable to healthcare professionals in medical imaging classification [175,228]. Due to its feature as a component-wise universal approximator, it can be formulated in multiple ways, including feature extractors dependent on preprocessing and domain knowledge, classifiers for discrimination between groups, or regressors for the prediction of scenarios. Neural networks can also be used in AD knowledge discovery as the feature representation extracted by neural networks might contain information that is counter-intuitive to human understanding. This review outlines the fundamental techniques of deep learning and the main categories of current approaches to various challenges. As a machine learning sub-branch, deep learning approaches can be categorized into two main categories: unsupervised learning and supervised learning.

## 6. Unsupervised Learning

Unsupervised learning extracts inferences without ground truth categorization of the provided data samples or labels, while supervised approaches require data sample and label pairs. In deep learning, no architecture is strictly supervised or unsupervised if we decompose them into their base components, e.g., feature extraction and classification components of convolutional neural networks. In this survey, the distinction is made based on the relationship between the optimization target of the main neural network or framework and ground truth labels. Unsupervised learning methods will be summarized in this section, while supervised learning methods will be summarized in Section 7.

### 6.1. Autoencoder (AE)

Autoencoders are a type of artificial neural network designed to learn efficient data representation. The classical application of autoencoders is an unsupervised learning method with two main components: the encoder $f_e$, and the decoder $f_d$. The encoder is a neural network designed to map the input to a latent feature representation, while the decoder is a mirror image of the encoder designed to reconstruct the original input from the compressed representation, i.e.,

$$x' = f_d(f_e(x)),\tag{3}$$

where $x'$ is the reconstructed input, and $f_e(x) = z$, is the latent representation. AE can obtain efficient data representations in an alternative dimension by minimizing a reconstruction loss, e.g., squared errors:

$$L\left(x, x'\right) = ||x - x'||^2\tag{4}$$

The original AE consists of fully connected layers, while a stacked autoencoder consist of multiple layers within the encoder and decoder to allow extraction of high-level representations. This structure can be directly applied to train on extracted features such as the ROI features detailed in Section 4.4. In a previous study, structural features of ROI were combined with texture features extracted from Fractal Brownian Motion co-occurrence matrices [229]. Since the AE is unsupervised, a supervised neural network component is attached after training to enable classification or regression. This component commonly consists of fully connected layers (FCL) and activations. Fine-tuning by re-training the network with the supervised component is often applied to achieve better performance.

Greedy layer-wise training can be applied since the encoder and decoder have similar structures regardless of the number of stacked layers. In this training protocol, layers are continuously added to the encoder and decoder and retrained for hierarchical representation. Liu et al. [230] integrated this protocol with multi-modal fusion to improve multi-class classification with MCI sub-types to 66.47% ACC with 86.98% specificity. The same AE also achieved higher performance for binary classification tasks of AD vs. HC and MCI vs. HC. Another commonly applied method to improve AE performance is using sparsity constraints on the parameters. The constraint can be applied through $l_1$-regularization or Kullback–Leibler divergence [231–233] for the model to learn with limited neurons during training instances and thereby reduce overfitting. For the classification between HC and MCI, Ju et al. [234] applied sparsity-constrained AE with functional connection matrices between ROIs in fMRI data. The sparsity constraint AE achieved a classification ACC of 86.47% with an AUC of 0.9164, over 20% higher than the machine learning counterparts of SVM, LDA, and LR. Apart from the training protocol and parameter constraint, some methods moderate the input and output of AE. Denoising AE reformulates the original reconstruction problem of AE to a denoising problem with the introduction of isotropic Gaussian noise.

$$x' = f_d(f_e(x + N(0, 1)))\tag{5}$$

Ithapu et al. [235] utilized this AE variant for feature extraction to construct a quantified marker for sample enrichment. Bhatkoti and Paul [236] applied a k-sparse autoencoder where only the neurons corresponding to the k-largest activations in the output are activated for backpropagation. These studies are representative of innovations in the application and enhancement of the original autoencoder.

The structure of neural networks in the encoder and decoder is not limited to MLP; the convolutional structure is also common among AD-related applications of AE. A study has applied 1D convolutional-AE to derive vector representations of longitudinal EHR data, where the 1D convolution operations act as temporal filters to obtain information on patient history [110]. Similarly, more sophisticated convolutional structures can also be used in the encoder and decoder architecture. Oh et al. [237] applied a convolutional AE with Inception modules, which are groups of layers consisting of multiple parallel filters. The standardized structure of AE makes it adaptable to any input dimension by configuring the encoder and decoder structure. Hosseini-Asl et al. [238], and Oh, Chung, Kim, Kim and Oh [237] applied 3D convolutional autoencoders to compress the representations of 3D MRI, while Er and Goularas [239] applied AE as an unsupervised component of the feature extraction process. AE can also be implemented as a pre-training technique, where after training, fully connected layers are added to the compressed layer of the encoder and used for supervised learning [240].

Apart from structural adjustments to the encoder and decoder layers, a probabilistic variation of AE also exists. These AE are known as variational autoencoders (VAE). For VAE, a single sample of available data $x_i$ can be interpreted as a random sample from the true distribution of data $p^{'}$, while the encoder can be represented as $q(z|x)$, an approximation to the true marginal distribution of $p(x|z)$. The loss function is, therefore,

$$L = L_1\left(x, x^{'}\right) + L_{KL}(q(z|x), p(z)), \tag{6}$$

where $L_1$ is the reconstruction loss and $L_2$ is the Kullback–Leibler divergence, which regularizes the VAE and enforces the Gaussian prior $p(z) = N(0, 1)$. Through this adjustment, AE learns latent variable distributions instead of representations [241]. A more intuitive formulation is as follows:

$$\mu = f_{h_1}(f_e(x)) \text{ and } \sigma = f_{h_2}(f_e(x)), \tag{7}$$

where $f_{h_1}$ and $f_{h_2}$ represent the mapping to two independent neural network layers representing $\mu$ and $\sigma$, the set of mean and variance of the latent distributions. The latent representation can be sampled through reparameterization,

$$z = \mu + \sigma \varepsilon \text{ where } \varepsilon \sim N(0, 1), \tag{8}$$

and decoded to reconstruct the input $x^{'} = f_d(z)$. Variational autoencoder has recently been applied to extract latent distributions of eMCI from high-dimensional brain functional networks [242] and provide risk analysis for AD progression [243]. Instead of a single set of latent distributions, a hierarchy of latent distributions can also be learned using ladder VAE. This variant of VAE was applied by Biffi et al. [244] to model HC and AD hippocampal segmentation populations, where latent distribution-generated segmentations for AD showed apparent atrophy compared with HC. By learning latent distributions, new data can be sampled from these distributions to generate new samples. From this perspective, VAE can be considered a generative model and is introduced in the following subsection. The fundamental autoencoder structures are shown in Figure 5.

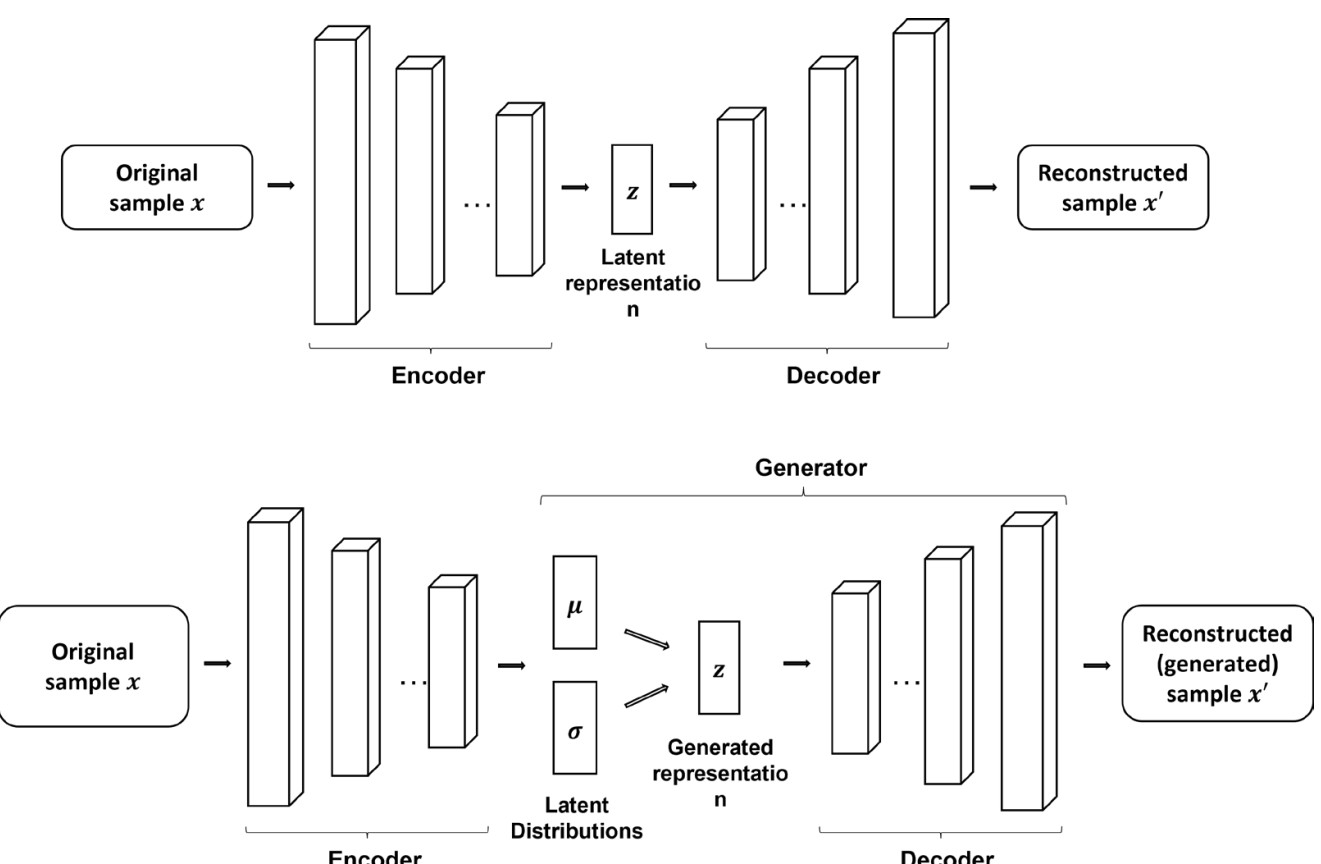

**Figure 5.** Fundamental autoencoder structures. The top figure represents a stacked 2D autoencoder, where each block represents a convolutional layer composed of a bank of convolutional filters (represented by the rectangular columns). The bottom represents a VAE, where instead of the latent representation, the encoder generates latent distributions represented by mean $\mu$ and variance $\sigma$, which are then used to generate representations. The convolution operation can be replaced by fully connected layers or complex modules, e.g., the Inception module.

*6.2. Generative Models*

Generative methods are a form of unsupervised learning that requires the model to recreate new data to supplement an existing data distribution. Variational autoencoders and RBM mentioned in the previous sections are both generative models. Another popular generative method is the construction of generative adversarial networks (GANs), where two or more neural networks compete in a zero-sum game. Classical GAN includes a generative neural network $G$ used to generate dummy data and a discriminator neural network $D$ to determine whether a sample is generated. The generator generates fake images $x' = G(\epsilon)$ from noise $\epsilon$. The generated sample belongs to the generated data distribution $x' \in p_g$. The discriminator attempts to discriminate between generated images $x'$ and real images, $x \in p_r(x)$. The competition between the generator and discriminator can be formulated through their loss function

$$L = \mathbb{E}_{G(\epsilon)\sim p_g}\{\log[1 - D(G(\epsilon))]\} + \mathbb{E}_{x\sim p_r}\{\log[D(x)]\}, \tag{9}$$

where the objective is to minimize $G$ and maximize $D$ [245]. GAN is widely used for medical image synthesis, reconstruction, segmentation, and classification [246]. Islam and Zhang [247] applied a convolutional GAN to generate synthetic PET images for AD, NC, and MCI. The GAN model generated images with a mean PSNR of 32.83 and a mean SSIM of 77.48. The generated data were then classified using a 2D CNN, which achieved 71.45% ACC. This performance drop illustrates the difficulty in synthesizing quality syn-

thetic images for training. A similar framework was proposed with shared feature maps between the generator and discriminator. With transfer learning, the framework achieved 0.713 AUC for SCD-conversion prediction [248]. Roychowdhury and Roychowdhury [249] implemented a conditional GAN, where the discriminator and generator are conditioned by labels $y$,

$$L = \mathrm{E}_{G(\epsilon) \sim p_g} \{\log[1 - D(G(\epsilon|y))]\} + \mathrm{E}_{x \sim p_r} \{\log[D(x|y)]\}, \tag{10}$$

The conditional GAN was applied to generate longitudinal MRI data by generating and overlaying cortical ribbon images. The generated data provide a potential disease progression model of MCI to AD conversion and brain atrophy. The study showed that the modeled fractal dimension of the cortical image decreases over time. Baumgartner et al. [250] applied an unsupervised Wasserstein GAN, where a K-Lipschitz constraint critic function $C$ replaces the supervised discriminator. The loss of this model can be formulated as:

$$L = \mathrm{E}_{x \sim p(y=1)} \{\log[C(x + M(x))]\} + \mathrm{E}_{x \sim p(y=0)} \{\log[C(x)]\}, \tag{11}$$

where $D$ is a set of 1-Lipschitz functions, and $M$ is a map generator function that uses existing images $x$ to generate new images $x' = x + M(x)$. An additional regularization component $L_M = \|M(x)\|_1$ is also added to the overall loss function to constrain the map $M$ for minimum change to the original image $x$. In the study, $M$ is modeled by a 3D U-Net segmentation model. The modified WGAN generated disease effect maps similar to human observations for MRI images of MCI-converted AD. An alternative application of Wasserstein GAN with additional boundary equilibrium constraints was applied by Kim et al. [251]. This study extracted latent representations from autoencoder structure discriminators for classification with FCL and SVM. For AD vs. HC, the model achieved an ACC of 95.14% with an AUC of 0.98. A subsequent study by Rachmadi et al. [252] built upon the Wasserstein GAN structure with an additional critic function $C_2$. The loss function corresponding to this additional component is:

$$L_{c_2} = \mathrm{E}_{x_1, x_0 \sim p_1, p_0}[C_2(x_1 - x_0)] - \mathrm{E}_{x_0 \sim p_0}[C_2(M(x_0))], \tag{12}$$

where $x_0$ and $x_1$ are baseline and follow-up images, respectively. Apart from using the original critic $C$ to discriminate between real and fake images, the new critic $C_2$ is also applied to discriminate between real disease evolution maps $x_1 - x_0$ and generated maps $M(x_0)$. The inclusion of $C_2$ reformulates the generation of dummy scans to the generation of longitudinal evolution maps. Though this study was applied in monitoring the evolution of white matter hyperintensities in cerebral small vessel disease, the same concept and technique can be migrated to data on Alzheimer's and related diseases [252]. Example GANs are illustrated in Figure 6. Apart from GAN, another type of innovative generative model is invertible neural networks (INN), which create invertible mappings with exact likelihood. Sun et al. [253] used two INN to extract the latent space of MRI and PET data and map them to each other for modality conversion. Conditional INNs, based on the conditional probability of latent space and combined with recurrent neural networks (RNN), were also used to generate longitudinal AD samples [176].

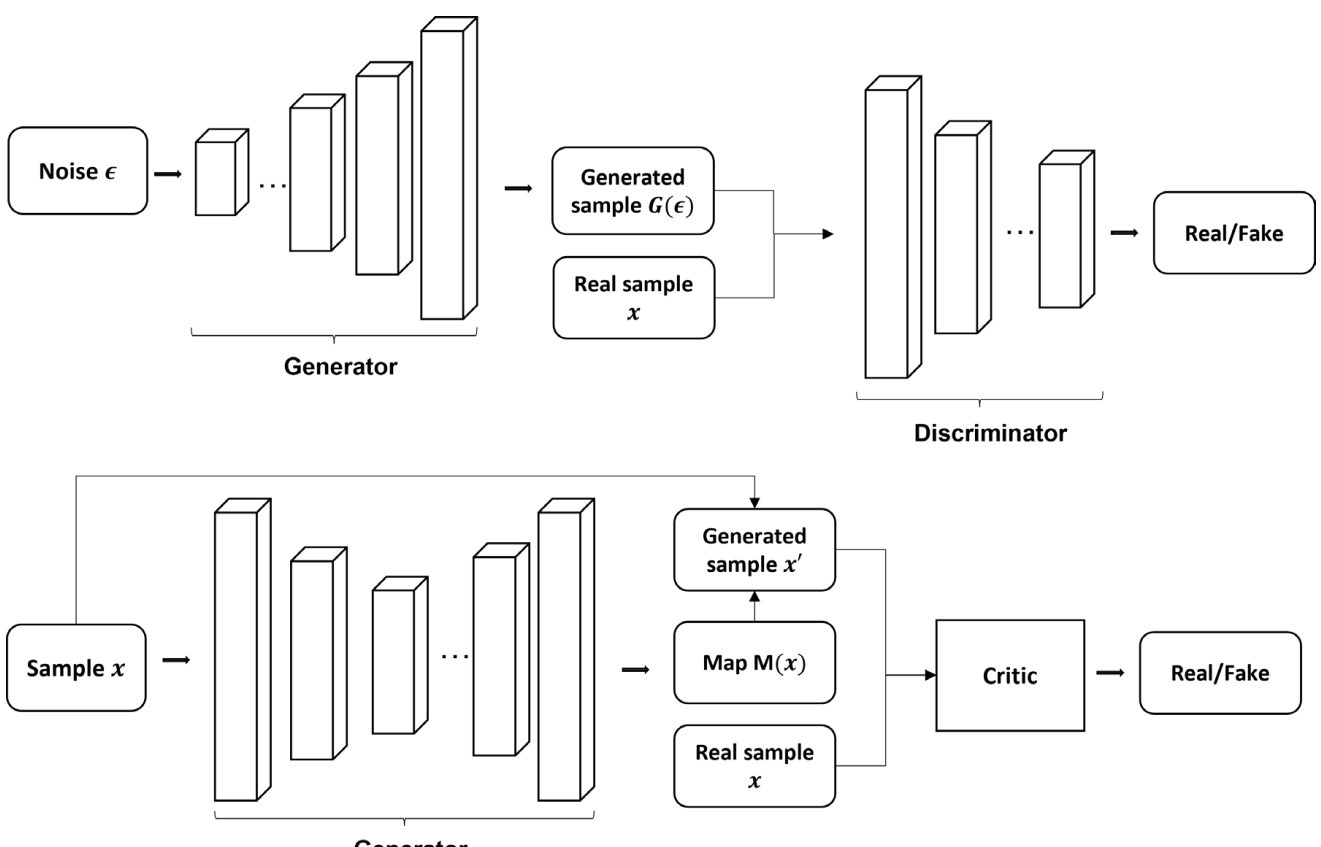

**Figure 6.** Example generative adversarial networks. The top figure is an example vanilla 3D convolutional generative adversarial network. The bottom figure shows the basic schematics of the modified Wasserstein GAN [250,252]. The structure of each generator and discriminator component can be modified for different neural network architectures.

*6.3. Restricted Boltzmann Machine (RBM) and Other Unsupervised Methods*

Apart from GAN and AE, numerous unsupervised methods have been applied for AD and related diseases. A well-known category is the restricted Boltzmann machine (RBM). An RBM is a generative network with a bipartite graph used to extract the probability distributions of the input data. RBM consists of two symmetrically-linked layers containing the visible and hidden units, respectively. The units, or neurons, within each layer are not connected. Similar to autoencoders, RBMs encode the input data through the forward pass while reconstructing input data through its backward pass. Two sets of biases for the two different passes aid this process. As an unsupervised method, RBM can also be used for feature extraction. Li et al. [254] applied multiple RBMs to initialize multiple hidden layers one at a time, while Suk et al. [255] combined RBM with the autoencoder learning module by combining layer-wise learning with greedy optimization. Conditional RBM has been applied as a statistical model for unsupervised progression forecasting of MCI, achieving ADAS-Cog13 prediction performance compared with supervised methods [256]. A deep belief network (DBN) is a neural network architecture comprising stacked RMBs. The basic structure of a DBN is shown in Figure 7. A DBN allows a backward pass of generative weights from the extracted feature to the input, making it more robust to noise. However, the layer-by-layer learning procedure for DBN can be computationally expensive. Suk, Lee, Shen and Initiative [166] applied a combination of MLP and DBM for feature extraction from multiple modalities.

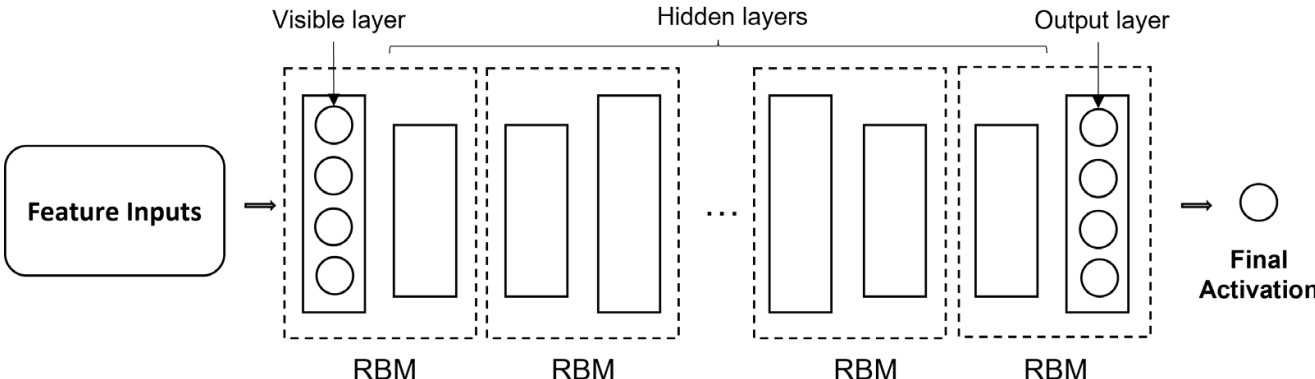

**Figure 7.** The basic structure of a deep belief network (DBN) consists of multiple restricted Boltzmann machines (RBM).

More recent studies in unsupervised learning applied show great diversity. Razavi et al. [257] applied sparse filtering as an unsupervised pre-training strategy for a 2D CNN. Sparse filtering is an easily applicable pre-training method where a neural network is first trained to output in a specified feature dimension. In this study, the cost function of classification is replaced by minimizing the sparsity of $l_2$-normalized features of specified dimensions. Bi et al. [258] combined a CNN with PCA-generated filters and k-means clustering for a fully unsupervised framework for clustering MRI of AD, MCI, and NC. Wang, Xin, Wang, Gu, Zhao and Qian [184] hierarchically applied extreme learning machines for unsupervised feature representation extraction. Extreme learning machines are a variant of feedforward neural networks that applies the Moore–Penrose generalized inverse instead of gradient-based backpropagation. Majumdar and Singhal [259] applied deep dictionary input while using noisy inputs, such as denoising autoencoders, for categorical classification, while Cheng et al. [260] utilized a U-net-based CNN with rigid alignment for cortical surface registration of MRI images.

## 7. Supervised and Semi-Supervised Learning

Supervised learning involves the use of known labels. In this study, we focus on the use of neural networks to map inputs to definite outputs. This section first introduces architecture classes, such as convolutional and recurrent neural networks, in Sections 7.1 and 7.2. We then present recent advances in transfer learning, ensemble learning, and multimodal fusion in Sections 8.1–8.3. Finally, we introduce the most recent developments in graph and geometric neural networks.

### 7.1. Convolutional Neural Networks (CNN)

The innovation of convolutional neural networks (CNN), especially the development of the AlexNet [261,262] by Krizhevsky et al. [263], validated neural networks as practical universal approximators with layer-wise feature propagation. In CNN, the dense connections of MLPs are replaced with kernel convolutions:

$$f(\boldsymbol{x})_{m,n} = \sigma\left(\sum_i^H \sum_j^W \sum_c^C x_{i,j,c} K_{m+i-1,\,n+j-1,c}\right), \tag{13}$$

where $K$ is the convolutional kernel; $\sigma$ is a non-linear activation function; and $H$, $W$, and $C$ represent the dimensions for height, width, and channel of the input. CNN allows for parameter-efficient hierarchical feature extraction. Besides the reduced computational requirements, CNN has translational invariance and can retain spatial information, making it particularly suitable for neuroimaging data. The effectiveness of CNN is evident in their broad application, both as an independent model and as network components [264].

A typical CNN consists of several convolutional layers followed by non-linear activations. The non-linearity provides the basis for learning through backpropagation. Commonly used activation functions include the rectified linear unit (ReLU), $\sigma = \max(0, x)$, the hyperbolic tangent, $\sigma = \tanh x$, and sigmoid functions, $\sigma = (1 + e^{-x})^{-1}$. Recent new activation functions such as leaky-ReLU and parametric-ReLU are also seen in the reviewed literature [177,265].

Pooling, the downsampling of feature maps through an average or maximum filter approach, is also often applied. Batch normalization, where each mini-batch of data is standardized, is also commonly applied after convolution. A combination of the above procedures forms a convolution block, and a typical CNN comprises multiple convolution blocks. These blocks are often followed by a few fully connected layers and a Softmax activation for classification or a linear activation for regression. The theoretical foundations of CNN can be understood through the decomposition of tensors [266], while in this paper, we will focus on practical applications of CNN for AD-related tasks. The following subsections will provide a summarized introduction to 2D and 3D CNN focusing on recent applications, while more detail can be found in previous reviews [89,96].

### 7.1.1. 2D-CNN

The original CNN was designed for computer vision pattern recognition of 2D images, allowing an easy application for 2D neuroimaging data. A basic 2D-CNN is shown in Figure 8. Aderghal, Benois-Pineau and Afdel [189] used a two-layer CNN with ReLU and max-pooling of 2D+ $\varepsilon$ images that project slices from the sagittal, coronal, and axial slices into a three-channel 2D image. Alternatively, when 2D slices are available from multiple planes of a 3D image, an individual 2D-CNN can be used for each image and then ensembled. Neural network depth is associated with an increase in performance. Wang, Phillips, Sui, Liu, Yang and Cheng [265] proposed a deeper eight-layer CNN with leaky rectified learn units to classify single-slice MRI images, while a similar CNN was applied for the classification of Florbetaben-18 PET images [177]. Tang et al. [267] used a CNN model to identify amyloid plaques in AD histology slides. Similar to the aforementioned 2D CNN, the neural network consists of alternating layers of 2D convolution and max-pooling, followed by fully connected layers with ReLU activation and a Softmax activation to produce classification outputs. The CNN model showed excellent performance in the classification of amyloid plaques with an AUC of 0.993. The current state-of-art 2D CNN models are also mostly developed for natural image classification, though these models are easily applicable for 2D AD-related data. The availability of pre-trained state-of-art models provides the basis of transfer learning, as summarized in Section 8.1.

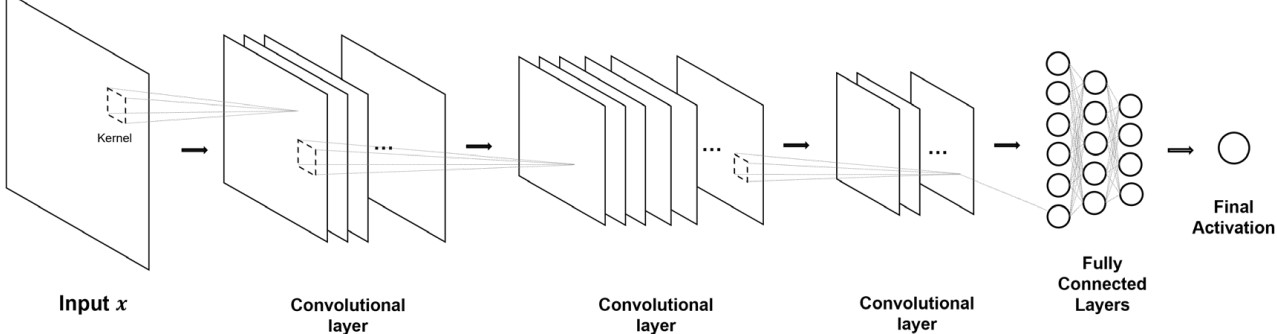

**Figure 8.** Example 2D-CNN. This architecture provides the foundation for 2D convolutional architectures. Square slices in this figure represent channel-wise feature maps after convolution.

Due to the two dimension limit, data with multiple slices are either treated as independent or similar. 2D-CNN can also be applied to 1D data, including using the Hilbert space-filling curve to transform 1D cognitive assessment data to 2D [268] or using the time-series data of multi-channel EEG as a 2D matrix [269]. Though limited by dimensionality, 2D-CNN can be more practical in real-world application and deployment, as the data used in clinical practice are often 2D or lacks enough slices to construct the high-dimensional 3D T1-weighted MRI predominately used for medical research. The 3D neuroimaging data in open libraries such as ADNI and OASIS are often processed to obtain 2D slices or patches, as mentioned in Section 4. To retain 3D spatial information, 2D slices or patches from the sagittal, coronal, and axial views are often extracted for multi-view networks [270]. The lower dimensionality of 2D-CNN also makes it suitable for adaptation for 1D data, e.g., Alavi et al. [271] utilized the triplet architecture of face recognition and the Siamese one-shot learning model for automated live comparative analysis of RNA-seq data from GEO.

### 7.1.2. 3D-CNN

3D-CNN is inherently the same as 2D-CNN apart from an additional dimensionality in all components, including the convolutional kernel. The additional dimension provides 3D-CNN with better spatial information than 2D-CNN as the latter is inherently limited by kernel dimensionality and is, therefore, unable to efficiently capture the spatial information between slices. A basic 3D-CNN is shown in Figure 9. Similar to fundamental 2D-CNN models, Islam and Zhang [272] used a 3D-CNN composed of four 3D convolutional layers with FCL and Softmax with T1-weighted MRI, while Duc et al. [273] applied a similar CNN with rs-fMRI functional networks. A simple two-block 3D-CNN applied by Basaia et al. [274] showed either comparable or better performance than 2D-CNN in binary classification with AD, NC, and various MCI subtypes.

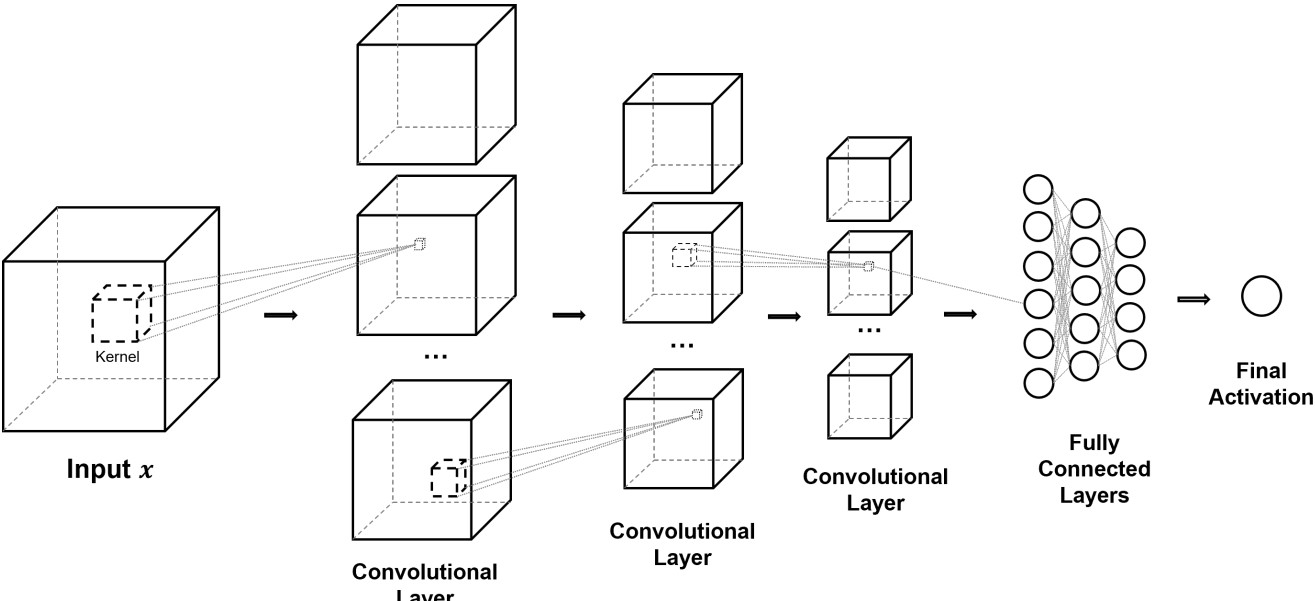

**Figure 9.** Basic 3D CNN architecture. 3D images and patches from Figure 4 can be used as inputs for this architecture. Individual blocks within a convolutional layer represent channel-wise feature maps after convolution. Modifications such as identity mapping and dense connectivity can be applied with an additional dimension of height. Fully connected layers can be replaced with global average pooling for a fully convolutional neural network, while the final activation can be modified for classification, regression, or additional structure can be applied for alternative tasks such as semantic segmentation.

With the similarity between 2D and 3D CNN, high-performing architectures in two dimensions can easily be adapted to three dimensions; Basaia, Agosta, Wagner, Canu, Magnani, Santangelo, Filippi and Initiative [274] and Qiu et al. [275] both implemented 3D versions of an all convolutional neural network for the classification of AD and MCI, where the FCL + Softmax classification component is replaced with a CNN with a channel number corresponding to the number of categories and global pooling of each channel. A similar application of an all convolutional CNN was applied by Choi et al. [276] for MCI conversion prediction. Unsupervised pre-training has also been tested by Hosseini-Asl, Keynton and El-Baz [238] and Martinez-Murcia et al. [277] with 3D convolutional autoencoders, while features extracted by 3D-CNN have also been used as input for sparse autoencoders [278]. Ge et al. [279] combined a U-Net-structured 3D-CNN for multi-scale feature extraction with XG-Boost feature selection. State-of-art architectures for 2D-CNN have also been adapted to 3D, e.g., a 3D architecture for the Inception-v4 network [280]. Liu et al. [281] used 3D-AlexNet and 3D-ResNet as comparative models. Wang et al. [282] also proposed a probability-based ensemble of densely connected neural networks with 3D kernels to maximize network information flow. This study also revealed ensemble learning as a potential approach to higher performance, which is detailed in Section 8.2.

The additional dimensionality of 3D does not restrict input to the spatial domain. An example is dynamic functional connectivity networks, which are 2D representations of brain ROIs' changes in blood oxygen level-dependent (BOLD) signals over time. For input of FCNs, the 3D-CNN obtains an additional temporal dimension in addition to the 2D spatial representation. With convolution along the temporal dimension, the neural network combines temporal and spatial connectivity to form more dynamic FCNs that can characterize time-dependent interactions considering the different contributions of time points [186].

The additional dimensionality of 3D-CNN corresponds to a significantly higher number of parameters within the model and higher computational cost. In order to reduce the computational cost, Spasov et al. [283] applied parameter-efficient 3D separable convolution, where the original 3D convolution is divided into depth-wise convolution and $1 \times 1$ point-wise convolution. Liu, Yadav, Fernandez-Granda and Razavian [281] performed comparative and ablation experiments and found that instance normalization can generalize better than batch normalization. This study also found that early spatial downsampling negatively impacts model performance, indicating that wider CNN architecture is more beneficial than additional layers and that smaller initial kernel sizes are ideal.

Liu, Cheng, Wang, Wang and Initiative [170] proposed the combined use of an ensemble 3D-CNN and 2D-CNN in a sequential manner, where the 3D-CNN captures spatial correlations with the 3D input. An ensemble of cascading 3D-CNN-generated feature maps is used as input for 2D-CNNs. While most of the studies above focus on categorical classification, disease progression predictions, and prediction of clinical measures, some deep learning studies were applied for different purposes, e.g., segmentation and image processing, which is potentially valuable for future studies in Alzheimer's and related diseases. Yang et al. [284] proposed a 3D-CNN with residual learning architecture for hippocampal segmentation that is significantly more efficient than conventional algorithms. Pang et al. [285] combined a semi-supervised autoencoder with local linear mapping. With the development and availability of more powerful hardware in the past decade, the 3D convolutional neural network has become increasingly popular amongst applications within the reviewed literature.

### 7.2. Recurrent Neural Networks (RNN)

Longitudinal data of AD provide multiple data instances of a subject, allowing us to find ground truth for MCI conversion and time-to-conversion. However, the temporal nature of a series of instances is often not explored in DNN and CNN architecture. Recurrent neural networks incorporate the temporal domain through adaption to a sequence of input with time-varying activation and sequential synapse-like structure. The fundamental concept was formulated by Goodfellow et al. [227] as:

$$h^t = f'\left(h^{t-1}, x^t\right), \tag{14}$$

where $h^t$ and $x^t$ represent the state and input at time step $t$. The state can be unfolded with respect to the past sequence:

$$h^t = g^t\left(x^1, x^2, \ldots, x^t\right), \tag{15}$$

where $g^t$ is a function. This property of the vanilla RNN allows $f$ to learn on all time steps and sequence lengths. Second-order RNNs consist of more complex neurons with memory components such as long short-term memory. LSTM is composed of a memory cell and three gates. The gates can be formulated as:

$$g_c{}^t = \sigma\left(b_c + U_c x^t + W_c h^{t-1}\right), \tag{16}$$

where for each gate $c$, $\sigma$ represents the activation, and $W_c$ and $U_c$ represent the recurrent weight and input weight matrices, respectively. The cell and update protocol can be formulated as:

$$s^t = g_f^t s^{t-1} + g_e^t \cdot \sigma\left(b + U x^t + W h^{t-1}\right), \tag{17}$$

$$h^t = \tanh\left(s^t\right) \cdot g_o^t, \tag{18}$$

where $g_f$ is the forget gate, $g_e$ is the external input gate and $g_o$ is the output gate. The recurrent weight and input weight of the memory cell are represented as $U$ and $W$, respectively.

LSTM has been applied to brain network graph matrices to extract adjacent positional features from fMRI data; the combination of LSTM and extreme learning machine (ELM) showed a slight improvement over a CNN-ELM model in classification tasks [185]. Gated recurrent unit (GRU) is another gated-RNN structure that shares a similar structure with LSTM but does not contain the forget gate. Therefore, it contains a lower number of parameters and is more suitable for capturing long-term temporal patterns. GRU has been used for classification with temporal clustering of actigraphy time-series obtained through the monitoring of activity for NC, MCI, and AD subjects. In this application, features extracted with CNN and Toeplitz inverse covariance-based clustering were combined and fed into the recurrent neural network [286].

Bi-directional GRU (BGRU) is a GRU variation that can process input both forwards and backwards. It has been applied in a similar manner to MLP and CNN-extracted features in multiple studies [287,288]. Apart from its use as a classification component to replace traditional MLP or machine learning classifiers, RNN can also be utilized for structural data. One study has combined CNN and RNN by inputting a series of 2D slices from 3D scans to capture spatial features; the CNN component captures features within single slices, while the BGRU structures obtain a time-series of CNN-extracted features to extract inter-slice features, which are then used as input for an MLP classifier component [289]. Similarly, instead of features extracted from slice-level data, LSTM architecture variants have been modified to suit 3D structural data, e.g., 3D convolutional LSTM to encode representations extracted by a 3D-CNN [290].

In the range of applications of RNN in deep learning, a key characteristic that stands out is its ability to deal with temporal data. Therefore, one focus of interest is the combination of spatial and temporal information. Wang et al. [291] combined the two types of information for fMRI data through the parallel implementation of multiple LSTM on features corresponding to multiple time series of ROI BOLD signals and the use of convolutional components for time-series segments. While most studies formulate the MCI prediction problem as a classification task [292], one study has used features extracted from an LSTM-based autoencoder for prognosis modeling with a Cox regression model [293]. For time-series or sequential data, sample completeness is a significant challenge in practice. Due to the difficulty in longitudinal data collection, many datasets have missing or delayed collection time points. On top of classical data imputation, Nguyen et al. [294] utilized their proposed minimal RNN model to impute missing data by filling it with model predictions. This study achieved exceptional results in the TADPOLE longitudinal challenge for the 6-year predictions of ADAS-Cog13 and ventricular volume. These studies have shown the effectiveness of RNNs in the temporal modeling of AD and related diseases. With the incrementally increasing amount of longitudinal data collected across various projects, they will significantly impact the direction of the deep learning approach.

### 7.3. Graph and Geometric Neural Networks (GNNs)

The underlying assumption for conventional neural networks is that the latent distribution of data lies in the Euclidean domain. Graph and geometric neural networks are a branch of deep learning designed for data in the non-Euclidean domain [295], e.g., genetic pathways, brain manifolds, and functional networks. Before the development of GNN, the dominant deep learning approach to graph data was graph kernel methods, where a kernel function is used to map the graph into vector space as input for neural networks. Studies have utilized this method by calculating brain function networks represented as correlation matrices and then using them as input for various neural networks [185,296]. Therefore, most brain function network studies can be considered graph kernel methods. The pipeline to generate these vectors is deterministic, while GNN is learnable and is relatively less penalized by the curse of dimensionality with relational data [297].

There are many categories of GNN, but the most popular GNN in AD research are graph convolutional neural networks (GCNN), which can be sub-categorized into special and spatial GCNN. Song et al. [298] utilized GCNN for multi-class classification and verified a performance advantage over traditional machine learning classifiers with a low sample size. The GCNN was then applied to predict tau protein trajectory with a constraint on loss based on a physical model of tau protein spread in the brain [299]. Song et al. [300] proposed a GCN framework with similarity-aware receptive fields and adaptive adjacency matrices generated through pre-training for better prediction.

A major sub-category of GCNN is spatial GCNN, where we reformulate convolution operations onto the graph nodes to exploit their spatial relationships [301]. A simple formulation of this process is presented by Wu, Pan, Chen, Long, Zhang and Philip [297], where for each layer $k$,

$$h^k = \sigma\left(XW^k + \sum_{i=0}^{k-1} Ah^{k-1}\Theta^k\right), \tag{19}$$

where $A$ is an adjacency matrix that contains the connection information between graph nodes; $X$ is the feature matrix of the graph; and $W$ and $\Theta$ are matrices of learnable parameters. A key aspect of utilizing GNNs is the generation of graphs or manifolds, e.g., structural connectivity graphs derived from DTI [298] and hypersphere projections of brain-functional networks extracted from fMRI [302]. The graph generation component can also be incorporated into the neural network with embedding and attention-based mechanisms [303], while global attention mechanisms can also be used to build resilience against noise and variance [304].

Spectral GCNN redefines convolution operations to the Fourier domain through the eigendecomposition of graph Laplacian [305]. For a simplified channel-wise example, the spectral GCNN can be formulated as:

$$h' = \sigma\left(U\omega U^T h\right),\qquad(20)$$

where $\omega$ denotes the channel component of the filter, which contains trainable parameters, and $U$ are the eigenvalues of a normalized graph Laplacian. $L = U\Lambda U^T$ and $U^T x$ are equivalent to the Fourier transform of $x$ [297]. Wee et al. [306] generated graphs based on the cortical thickness of structural MRI and implemented a spectral GCNN for classification between disease stages. The model achieved 92% accuracy in predicting late MCI conversion to AD. Similarly, Zhao et al. [307] utilized a Cheby-GCN-based spectral GCNN with graphs constructed upon MCI functional connectivity networks, hardware, and gender information to predict MCI. Similar to the application of the attention mechanism with spatial-GCNN, Kazi et al. [308] combined spectral-GCNN with an attention module based on LSTM for personalized diagnosis. Huang and Chung [309] implemented Monte-Carlo dropout on a similar network structure for uncertainty estimation in the prediction of MCI conversion. Yu et al. [310] proposed a spectral-GCN framework that simulates random walks with parallel GCN layers and takes a combined input of structural connectivity from DTI and functional connectivity from fMRI. The model showed the difference in the structural connection between different disease stages and achieved 84~93% accuracy for binary classification tasks between NC, early MCI, and late MCI [310].

As an emerging field in deep learning, many AD-related studies focus on various other fields of interest in geometric neural networks, including geometric deep learning manifolds, e.g., Zhen et al. [311] implemented a dilated convolutional architecture designed for sequential manifold-valued data and the application of spectral-temporal neural networks for EEG and fMRI data to capture both spatial and temporal information [312,313]. Geometric and graph neural networks represent a more general structure than the rigid Euclidean domain of conventional neural networks. This property is more suitable for inherently non-Euclidean data and can facilitate better integration of a variety of data types. GNN is becoming a significant area of research for developing future neural networks in AD research.

### 7.4. Other Methods

Other methods include reinforcement learning, a topic of artificial intelligence research that branches apart from supervised or unsupervised learning. Instead of learning representations, reinforcement learning models focus on agents' actions within an environment. Tang, Uchendu, Wang, Dodge and Zhou [112] applied reinforcement learning with natural language processing techniques for an MCI screening dialogue agent. The reinforcement learning environment was set up with the Actor-Critic method, where a user simulator neural network generates new dialogue data. This set-up is very similar to GAN, but for GAN, the actor cannot affect the reward of the critic function [314]. While the perceptron units of neural networks simulate human brain neurons' fundamental function, it is an oversimplistic representation. Current research in deep learning has attempted to create neural networks based on more representative biological neurons. An example of this research field is spiking neural networks (SNN). Compared with the sequential nature of RNNs, SNNs are neural networks inherently temporal by design. Capecci et al. [315] provided a proof-of-concept with an SNN architecture using EEG data for the prediction of MCI conversion.

We summarize the literature mentioned in this section in Tables 3–5.

**Table 3.** Binary classification results of selected literature between AD, NC, and MCI.

| Study | Data Modalities | Number of Subjects | | | Classification ACC (%) | | Classification AUC | |
|---|---|---|---|---|---|---|---|---|
| | | AD | NC | MCI | AD vs. NC | MCI vs. NC | AD vs. NC | MCI vs. NC |
| Suk and Shen [316] | MRI, PET | 51 | 52 | 99 | 95.9 | 85 | - | - |
| Suk, Lee, Shen and Initiative [166] | MRI, PET | 93 | 101 | 204 | 95.35 | 85.67 | - | - |
| Liu, Liu, Cai, Che, Pujol, Kikinis, Feng and Fulham [230] | MRI, PET | 85 | 109 | 77 | 82.59 | 82.10 | - | - |
| Li, Tran, Thung, Ji, Shen and Li [254] | MRI, PET, CSF | 51 | 99 | 52 | 91.4 | 77.4 | - | - |
| Aderghal, Benois-Pineau and Afdel [189] | MRI | 188 | 228 | 399 | 69.53 | 91.41 | - | - |
| Suk et al. [317] | MRI | 186 | 393 | 226 | 91.02 | - | 0.927 | - |
| Majumdar and Singhal [259] | MRI, PET, CSF | 51 | 99 | 52 | 95.4 | 85.7 | - | - |
| Cui, Liu and Li [287] | MRI | 198 | 229 | - | 89.69 | - | 0.9214 | - |
| Shi et al. [318] | MRI, PET | 51 | 52 | 99 | 97.13 | 87.24 | 0.972 | 0.901 |
| Liu, Wang, Tang, Hu, Wu and Pan [210] | MRI | - | 303 | 83 | - | 90.9 | - | - |
| Lu et al. [319] | PET | 226 | 304 | 521 | 93.58 | - | - | - |
| Ning et al. [320] | MRI, Genetic | 138 | 225 | 358 | - | - | 0.992 | - |
| Liu, Cheng, Wang, Wang and Initiative [170] | MRI, PET | 93 | 100 | 204 | 93.26 | 74.34 | 0.957 | 0.802 |
| Ge, Qu, Gu and Jakola [279] | MRI | 193 | 139 | - | 93.53 | - | - | - |
| Ju, Hu and Li [234] | fMRI | - | 79 | 91 | - | 86.47 | - | 0.916 |
| Liu, Zhang, Adeli and Shen [191] | MRI | 227 | 249 | 390 | 93.7 | - | - | - |
| Islam and Zhang [272] | PET | 169 | 400 | 661 | 88.76 | - | - | - |
| Wen, Thibeau-Sutre, Diaz-Melo, Samper-González, Routier, Bottani, Dormont, Durrleman, Burgos and Colliot [89] | MRI | 336 | 330 | 787 | 87 [b] | - | - | - |
| Liu, Li, Yan, Wang, Ma, Shen, Xu and Initiative [169] | MRI | 97 | 119 | 233 | 88.9 | 76.2 | 0.925 | 0.775 |
| Lee et al. [321] | MRI | 198 | 229 | 374 | 92.75 | 89.22 | 0.980 | 0.957 |
| Lian, Liu, Zhang and Shen [192] | MRI | 358 | 205 | 2964 | 89.5 | - | 0.959 | - |
| Cui and Liu [187] | MRI | 192 | 223 | 396 | 92.29 | 74.64 | 0.75 | 0.797 |
| Martinez-Murcia, Ortiz, Gorriz, Ramirez and Castillo-Barnes [277] | MRI | 99 | 168 | 212 | 84.9 | - | - | - |
| Duc, Ryu, Qureshi, Choi, Lee and Lee [273] | fMRI | 133 | 198 | - | 85.3 [b] | - | - | - |

**Table 3.** *Cont.*

| Study | Data Modalities | Number of Subjects | | | Classification ACC (%) | | Classification AUC | |
|---|---|---|---|---|---|---|---|---|
| | | AD | NC | MCI | AD vs. NC | MCI vs. NC | AD vs. NC | MCI vs. NC |
| Kim, Lee, Lee, Oh, Yun and Yoo [251] | PET | 212 | 415 | - | 94.82 | - | 0.98 | - |
| Choi, Kim, Yoon, Lee, Lee and Initiative [276] | PET | 243 | 393 | 666 | | | 0.94 | - |
| Xia, Yue, Xu, Feng, Yang, Wang and Lei [290] | MRI | 198 | 299 | 408 | 94.19 | 79.01 | 0.96 | 0.88 |
| Ieracitano, Mammone, Hussain and Morabito [269] | EEG | 63 | 63 | 63 | 85.78 | 85.34 | - | - |
| Islam and Zhang [247] | PET | 98 | 105 | 208 | 71.45 | - | - | - |
| Qiu, Joshi, Miller, Xue, Zhou, Karjadi, Chang, Joshi, Dwyer and Zhu [275] | MRI, Demo, CA | 488 | 978 | - | 96.8 | - | 0.996 | - |
| Bashyam et al. [322] | MRI | 353 | 833 | 513 | 86 | 70.2 | 0.91 | 0.743 |
| Pan, Phan, Adel, Fossati, Gaidon, Wojak and Guedj [270] | PET | 237 | 242 | 526 | 93.13 | - | 0.9747 | - |

[b] Some studies applied balanced accuracy, where accuracy is weighted by categorical distribution.

**Table 4.** Results from selected studies of binary classification between cMCI and ncMCI.

| Study | Data Modalities | Time to Conversion | Number of Subjects | | ACC (%) | AUC |
|---|---|---|---|---|---|---|
| | | | cMCI | ncMCI | | |
| Suk, Lee, Shen and Initiative [166] | MRI, PET | | 78 | 128 | 75.92 | |
| Suk, Lee, Shen and Initiative [317] | MRI | 18 M | 167 | 226 | 74.82 | 0.754 |
| Ning, Chen, Sun, Hobel, Zhao, Matloff, Toga and Initiative [320] | MRI, Genetic | 24 M | 166 | 192 | | 0.835 |
| Lu, Popuri, Ding, Balachandar, Beg and Initiative [319] | PET | 36 M | 112 | 409 | 82.51 | |
| Cui and Liu [187] | MRI | | 165 | 231 | 74.64 | 0.777 |
| Spasov, Passamonti, Duggento, Liò, Toschi and Initiative [283] | MRI, Demo, CA, Genetic | 36 M | 181 | 228 | 86 | 0.925 |
| Lee, Choi, Kim, Suk and Initiative [321] | MRI | 18 M | 160 | 214 | 88.52 | |
| Choi, Kim, Yoon, Lee, Lee and Initiative [276] | PET | 36 M | 167 | 274 | | 0.82 |
| Lian, Liu, Zhang and Shen [192] | MRI | 36 M | 205 | 465 | 80.9 | 0.781 |
| Wen, Thibeau-Sutre, Diaz-Melo, Samper-González, Routier, Bottani, Dormont, Durrleman, Burgos and Colliot [89] | MRI | 36 M | 295 | 298 | 76 | |
| Er and Goularas [239] | MRI | | 125 | 169 | 87.2 | |
| Pan, Phan, Adel, Fossati, Gaidon, Wojak and Guedj [270] | PET | 36 M | 166 | 360 | 83.05 | 0.868 |

Abbreviations: M—months, cMCI—MCI converters, ncMCI—non-converters, w.r.t.—time of conversion.

**Table 5.** Multi-class classification results of selected studies.

| Study | Data Modalities | Classes | Accuracy |
|---|---|---|---|
| Liu, Liu, Cai, Che, Pujol, Kikinis, Feng and Fulham [230] | MRI, PET | AD, cMCI, ncMCI, NC | 64.07 |
| Dolph, Alam, Shboul, Samad and Iftekharuddin [229] | MRI | AD, MCI, NC | 58 |
| Shi, Zheng, Li, Zhang and Ying [318] | MRI, PET | AD, cMCI, ncMCI, NC | 57.00 |
| Liu, Zhang, Adeli and Shen [191] | MRI | AD, pMCI, sMCI, NC | 51.8 |
| Lee, Choi, Kim, Suk and Initiative [321] | MRI | AD, MCI, NC | 71.17 |
| Liu, Yadav, Fernandez-Granda and Razavian [281] | MRI | AD, MCI, NC | 70 |

## 8. Deep Learning Techniques

### 8.1. Transfer Learning

With the popularity of deep neural networks in medical diagnostic systems, common challenges exist in practical applications [323]. These challenges include the availability of medical data and relevant labels. Current computer vision success is based on the ImageNet [324] hierarchical database, which contains millions of annotated images [325]. However, medical images are much smaller in quantity and require expert knowledge for labeling [326–329]. A potential solution to this problem is transfer learning—the transfer of knowledge across domains [330]. In image classification applications, transfer learning is commonly implemented by transferring model structure, weights, or parameters for classification in different feature spaces and distributions. Neural networks with transferred parameters have been shown to outperform the same neural networks with randomized parameters in convergence and have lower requirements for complicated and time-consuming hyperparameter searches [331].

There are three types of transfer learning: (1) transfer from Image-Net pre-trained models, e.g., Ding, Sohn, Kawczynski, Trivedi, Harnish, Jenkins, Lituiev, Copeland, Aboian and Mari Aparici [175] used the pre-trained Inception-V3 for the classification of AD vs. MCI, Bae et al. [332] applied a modified Inception-V4 with custom preprocessing for classification of AD vs. CN, Lin et al. [333] used the pre-trained AlexNet with RVR for regression, and Chen, Stromer, Alabdalrahim, Schwab, Weih and Maier [148] selected pre-trained ResNet-152, VGG-16, and DenseNet-121 for screening and scoring of dementia using clock-drawing test images; (2) transfer from pre-trained networks for similar classification or prediction tasks, e.g., using a pre-trained network trained on one dataset for another dataset [90,281]; and (3) transfer from pre-trained networks used for different classification or prediction tasks, e.g., using an AD vs. NC pre-trained model for classification between pMCI and sMCI [192,237], or for MCI vs. NC [334].

Chen, Hsu, Yang, Tung, Luo, Liu, Hwang, Hwu and Tseng [163] transferred domain knowledge between different datasets for brain age prediction, while in a large-scale study, Bashyam, Erus, Doshi, Habes, Nasralah, Truelove-Hill, Srinivasan, Mamourian, Pomponio and Fan [322] transferred a model used for brain age prediction to AD vs. NC and MCI vs. NC. Similar domain transfer has also been applied for transfer from Alzheimer's disease to Parkinson's disease [276]. Transfer learning is also applied for other data types, including eye-tracking, where datasets such as MIT GazeCapture, which are unrelated to Alzheimer's or related diseases, can be utilized for gaze location estimation [116].

### 8.2. Ensemble Learning

Ensemble Learning in deep learning is the combination of multiple representations to achieve higher overall performance. Ensemble learning allows multiple representations and mitigates errors within individual neural networks [335]. These errors are not limited to misclassification but can also include underfitting or overfitting on training data. Underfitting occurs when the gradient descent is trapped in local minima, and the neural network fails to capture the underlying manifold of the training data. Conversely, overfitting occurs when irrelevant fluctuations in the training data are also captured by the neural network, resulting in lower generalization [336]. Ensemble learning has been widely applied in medical image classification [282,337] and the classification of AD and related diseases [338,339].

Ensemble learning can be performed at three levels: input, feature, and output. The input-level ensemble combines data prior to input into the neural network, e.g., the combination of adjacent slices of hippocampal data to construct mimic RGB channels [189] and the use of zero-masking for the fusion of concatenated MRI and PET inputs [340]. The feature-level ensemble combines features from patch-level, region-level, and subject-level sub-networks as input features for a classification module; Lian, Liu, Zhang and Shen [192] is an ideal example of a feature-level ensemble with hierarchical sub-networks at each level, where the outputs of each level are concatenated and used as input for the next level. The feature-level ensemble was also applied at individual feature levels, i.e., an ensemble of multi-scale patch-level sub-networks [319]. The output-level ensemble combines the predictions of component neural networks, e.g., through majority voting of prediction results [157]. Suk, Lee, Shen and Initiative [317] combined the outputs of multiple sparse regression models with varying regularization parameters for classification. Wang, Shen, Wang, Xiao, Deng, Wang and Zhao [282] utilized a probability-based fusion of softmax outputs from an ensemble of 3D-DenseNets. Apart from combining the outputs of neural networks, output-level ensembles also allow for the ensemble between neural networks and traditional machine learning classifiers [341]. A sub-category of ensemble learning is multi-view learning. Multi-view learning for AD neuroimaging is commonly linked to the 3D nature of available neuroimaging data and slice-based preprocessing, as described in Section 3.

Pan, Phan, Adel, Fossati, Gaidon, Wojak and Guedj [270] created a pyramid network of multiple CNN subnetworks with separable convolutions for each of the three views. The features were added for each view and concatenated for classification [270]. It is worth noting that although ensembling at all three levels is common amongst reviewed papers, there are only a few applications of the boosting method. Boosting is standard in machine learning applications [279]. In this method, individual components are trained sequentially in an adaptive manner. The ensemble with multiple modalities, also known as multi-modal fusion, is introduced in the subsequent section.

### 8.3. Multi-Modal Fusion

Individual modalities are fundamentally limited in their information content, e.g., genetic data cannot provide information on texture information of neuroimaging data, and MRI has good soft-tissue resolution but is not directly associated with Amyloid-$\beta$ protein depositions. Fusing information from different modalities can provide a more comprehensive perspective of AD and related diseases. Multi-modal fusion is a common practice in the reviewed literature due to the availability of multi-modal data for AD and related diseases. The standard fusion method is the feature-level ensemble mentioned in Section 8.2, where at a particular stage of the model architecture, features produced by modality-dependent components are fused through concatenation or merging [166,318,342,343].

Liu, Liu, Cai, Che, Pujol, Kikinis, Feng and Fulham [230] performed the zero-masking of a single modality for a stacked autoencoder, which took both MRI and PET as input and achieved the fusion of the two modalities through data reconstruction of one zero-masked modality with only the other modality. Demographics and genetic biomarkers

are often fused with neuroimaging data through the concatenation of features extracted with fully connected layers [191]. For studies with 1D data or engineered features, the direct fusion of multi-modal data through concatenation or combined processing is achievable, e.g., the fusion of cognitive scores, volumetric features, gene expression, and CSF biomarkers [254,344,345].

Multimodal fusion is often combined with multi-scale or multi-view learning, e.g., studies have trained individual neural networks for patches of different sizes by processing MRI and PET images, where the inputs of the neural networks are concatenated for classification [170,320,346]. Through intricately designed connections between 1D and 3D network structures, Senanayake et al. [347] fused MRI and neuropsychological data. Likewise, Spasov, Passamonti, Duggento, Liò, Toschi and Initiative [283] constructed a more extensive architecture with the fusion of additional demographic and APOE-e4 genetic markers to input and Jacobian of sMRI images. Using multi-modal data is expected to improve the performance of neural networks. However, multi-modal fusion can also be limited by the availability of multi-modal data, especially for longitudinal studies. A basic overview of the common multi-modal fusion methods is shown in Figure 10.

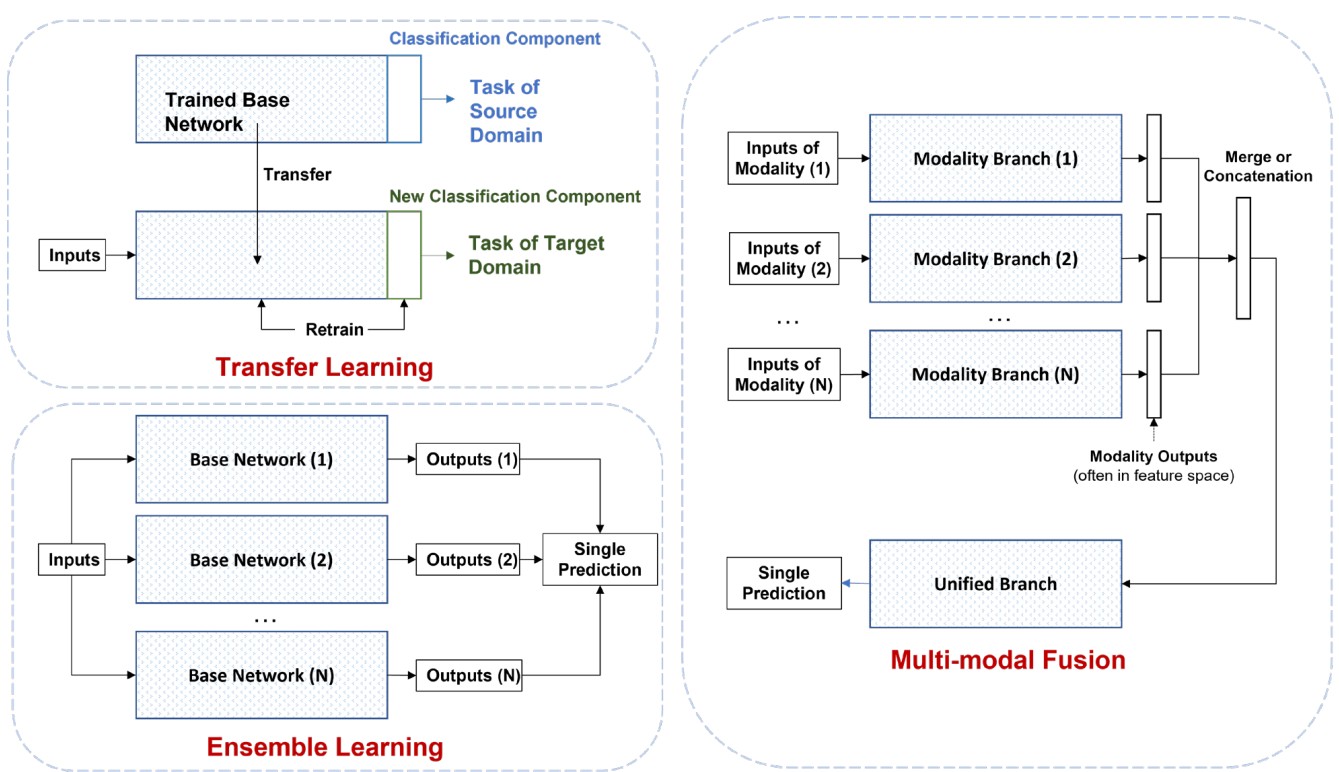

**Figure 10.** Overview of common multi-modal fusion methods applied in the field of AD, MCI and related diseases.

## 9. Training and Evaluation

The previous sections provide an overview of the myriad approaches to deep learning and relevant techniques applicable to its use for AD and related diseases. In general, two additional dependencies exist for any study/application of deep learning: how the method is trained and evaluated. Varying these two dependencies can generate wildly different results using the same approach and techniques. Different training and evaluation methods can also affect the interpretation and understanding of the results. In the following sections, we will first explore methods of evaluation in Section 9.1. This is the basis for an introduction to commonly applied training protocols in Section 9.2.

### 9.1. Evaluation Methods

#### 9.1.1. Hold-Out and Cross-Validation

Data-driven models commonly suffer from the effects of overfitting, where the model learns from the noise and variance within the training data. The fundamental evaluation method for any deep learning algorithm is an independent test set sampled from the same distribution as the training set. This method, also called hold-out, provides unbiased measures of evaluation. The test set is generally 20–50% of the entire cohort, split according to the number of individual subjects. The number of test subjects directly affects the approximate measures of generalization, including the Hoeffding inequality bounds.

Another commonly implemented method is cross-validation (CV) [348,349], a measure of model robustness. There are various types of cross-validation, including *k*-fold cross-validation [350], balanced cross-validation, randomized cross-validation, and leave-one-out cross-validation (LOOCV) [351,352].

The fundamental cross-validation method, also known as *k*-fold cross-validation, is performed by splitting the data into k equal folds of similar categorical distributions to the original cohort. For each of *k* rounds, a single fold is used as the validation set, while the remainder is used for training an independent model. Categorical imbalance can cause biased performance, whereas balanced cross-validation undersamples, or oversamples components of the cross-validation split to provide a balanced training or testing set. Randomized cross-validation does not adhere to the rigid *k*-folds; a random split is provided for each of the unlimited cross-validation rounds. LOOCV is a variation of *k*-fold cross-validation where $k = 1$; this is commonly used for data with a limited number of subjects.

#### 9.1.2. Metrics for Classification

The most common form of metric for classification is accuracy in predictions of defined labels. Apart from the basic classification accuracy, there are also sample-wise, subject-wise, and balanced accuracy, which weigh classification performance by categorical distribution. The measure of accuracy, or the correct classification rate, can be separated into a range of prediction measures, including true positive (TP), true negative (TN), false positive (FP), and false negative (FN). Similar metrics that measure various aspects of classification include the positive predictive value (precision), true positive rate (sensitivity), true negative rate (specificity), and the weighted combination of precision and sensitivity, the $F_1$ score. Simple variations of these metrics for binary classification:

$$\text{Accuracy} = \frac{\text{TP} + \text{TN}}{\text{TP} + \text{FP} + \text{TN} + \text{FN}}, \tag{21}$$

$$\text{Precision} = \frac{\text{TP}}{\text{TP} + \text{FP}}, \tag{22}$$

$$\text{Sensitivity} = \frac{\text{TP}}{\text{TP} + \text{FN}}, \tag{23}$$

$$\text{Specificity} = \frac{\text{TN}}{\text{TN} + \text{FP}}, \tag{24}$$

$$F_1 \text{ score} = \frac{2\text{TP}}{2\text{TP} + (\text{FP} + \text{FN})}, \tag{25}$$

Apart from these fundamental metrics, other metrics are also used for specific purposes. A common measure for imbalanced datasets is balanced accuracy (BAC), accuracy weighted by class distribution. Kim et al. [353] used Cohen's Kappa to provide a comparison between observed and random accuracy, while Son, Oh, Oh, Kim, Lee, Roh and Kim [177] and Mårtensson et al. [354] applied it to assess inter-method agreement. The receiver operating characteristic (ROC) curve visualizes the trade-off between specificity and sensitivity. The area under the curve (AUC) for the ROC curve indicates separability between binary class

probabilities. AUC is one of the most common metrics of classification in AD-related publications. Other metrics include the Gini-coefficient, derived from the AUC-ROC, and the Kolmogorov–Smirnov statistic which compares categorical probability distributions. These terms are commonly defined in binary terms but can be easily generalized to multi-class scenarios, providing a group of metrics for each category.

### 9.1.3. Metrics for Prediction

Since prediction can be formulated into classification problems, most classification metrics in Section 9.1.2 can be applied as prediction metrics. If the prediction problem is formulated as a regression, regression metrics can be used. These include measures of error such as the mean absolute error (MAE) and mean squared error (MSE):

$$\begin{cases} \text{MAE} = \frac{1}{n} \sum_i |y_i - y_i'| \\ \text{MSE} = \frac{1}{n} \sum_i (y_i - y_i')^2 \end{cases}, \tag{26}$$

where $n$ is the number of samples, $y_i$ are labels and $y_i'$ are predictions. Similar metrics include errors compared with simple predictors including the relative absolute error (RAE) and relative squared error (RSE),

$$\begin{cases} \text{RAE} = \frac{1}{n} \frac{\sum_i |y_i - y_i'|}{\sum_i |\overline{y_i} - y_i|} \\ \text{RSE} = \frac{1}{n} \frac{\sum_i (y_i - y_i')^2}{\sum_i (\overline{y_i} - y_i)^2} \end{cases}, \tag{27}$$

or the proportion of predictable variance, the coefficient of determination ($R^2$). Standard residual plots and residual analysis metrics can also be applied in this case. Since AD is a chronic medical condition, prediction can also be formulated as prognosis problems. Similar to the challenges in survival models, due to limitations in data collection from subjects suffering from AD or related diseases, there are cases of missing values or uncertain post-study outcomes. Metrics such as Harrell's C-index, or concordance index, take these 'censored' data into account by measuring the relationship between concordant and discordant pairs as follows:

$$\text{C index} = \frac{\sum_{i \neq j} 1_{t_i > t_j} \cdot 1_{\eta_i < \eta_j} \cdot \delta_j}{\sum_{i \neq j} 1_{t_i > t_j} \cdot \delta_j}, \tag{28}$$

where $t$ is time, $\eta$ represents risk scores, and $\delta \in (0, 1)$ are auxiliary variables indicating 'censorship'. Li et al. [355] measured concordance with other survival analysis measures, such as the Kaplan–Meier estimate. However, there is a lack of deep learning studies extending this approach to provide individualized risk models, such as the Cox proportional hazard model, with metrics such as the cumulative hazard. Moreover, there is a lack of deep learning-related research into the treatment effect of AD treatment methods where we measure individualized treatment effect (ITE) and C-for-benefit.

### 9.1.4. Other Metrics

A range of other metrics are also used for various purposes including data reconstruction and generation. Such metrics include the peak signal-to-noise ratio (PSNR):

$$\text{PSNR} = \frac{n \times \max(y)^2}{\text{MSE}}, \tag{29}$$

Another such metric is the structural similarity index (SSIM):

$$\text{SSIM}_{x,y} = \frac{\left(2\sigma_{x,y} + (K_2 L)^2\right)\left(2\mu_x \mu_y + (K_1 L)^2\right)}{\left(\mu_x^2 + \mu_y^2 + (K_1 L)^2\right)\left(\sigma_x^2 + \sigma_y^2 + (K_2 L)^2\right)}, \tag{30}$$

where $x$ and $y$ are two patches or images, $K_1$ and $K_2$ are constant values, $\mu_x$ is luminance, and $\sigma_x$ is contrast.

$$\mu_x = \frac{1}{n}\sum_i x_i \text{ and } \sigma_x = \sqrt{\frac{1}{N-1}\sum_i (x_i - \mu_x)^2}, \tag{31}$$

Both PSNR and SSIM are metrics of generative models. Normalized cross-correlation is another metric that is used to measure the quality of feature selection in the form of visual attributions:

$$\text{NCC} = \frac{1}{n}\sum_{x,y} y_{x,y} \cdot y'_{x,y'} \tag{32}$$

where $y'$ is the ground truth map for AD-affected regions. Segmentation metrics include the dice similarity coefficient, formulated in the same way as the F1-score in Section 9.1.2, where pixel-wise localization success is used instead of classification prediction.

### 9.1.5. Level of Evaluation

Data for AD and related diseases are inhomogeneous, with diverse data types and sources. Additional variance in preprocessing and processing data can provide significantly different inputs to the deep learning models. These differences give rise to problems in evaluation.

We can categorize two primary levels of evaluation: sample level and subject level. Sample-level evaluation is based on the model's performance in classifying or predicting data samples, while subject-level evaluation is based on individual subjects, e.g., AD or MCI patients. Sample-level evaluation occurs when multiple samples from the same subject are used in evaluation or when the data source does not indicate independence between samples. Subject-level evaluation can be based on either a single sample of data or multiple sample-level results; this provides a better representation in a real-world application.

With multiple data sources available for AD and related diseases, another level of evaluation has become more common in recent studies: validation with alternative datasets. This validation process involves using a trained model on a single dataset to provide outputs for data originating from an alternative source, e.g., a separate cohort or study. As the largest open library, data from ADNI is often used to train deep learning models, which are subsequently tested on data from other open libraries such as AIBL, OASIS [89], and private datasets [332].

### 9.1.6. Combination of Evaluation Methods

In current AD-related deep learning studies, a combination of evaluation techniques and metrics is often applied. A typical combination of evaluation techniques is cross-validation on the training set for hyper-parameter optimization and hold-out testing on the independent test set. The study-specific combinations are dependent on the overall objective of the deep learning model and training protocols applied to achieve this objective. Training protocols are discussed in detail in Section 9.2. In regard to classification, a combination of accuracy, sensitivity, and precision metrics is commonly measured for evaluation. Most MCI-conversion prediction problems are formulated as a classification based on a conversion time limit and share similar evaluation metrics.

### 9.1.7. Comparison and Ablation

Comparative studies provide insight into overall model performance, improvement, or limitation compared with the state-of-art studies of similar methods. In current AD-related deep learning literature, most studies apply comparative methods. These applications are often baseline machine learning or deep learning methods, such as SVM, Decision Tree, basic 2D-CNN, or variants of the proposed method. Most studies also contain a comparison of metrics from the literature. A comparison between models is commonly achieved by comparing the same or similar metrics under the assumption that the evaluation method and training protocols are similar. However, these comparisons can only serve as a rough performance evaluation due to differences in metrics definitions, data sampling, and processing methods. A major study to counter this problem of in-comparability and lack of reproducibility is the development of a standardized framework for machine learning algorithms, Clinica [90], and the extension of the framework for neural network evaluation [89].

With recent studies of increasingly advanced deep learning approaches, the performance increase between subsequent studies is small. For some studies, the performance gain in comparative models is within the approximated generalization error bounds. Therefore, instead of basic comparisons, some studies conducted statistical tests to validate performance gain. The Delong test [268,356], which produces a confidence interval and standard error of difference, can compare the AUC of comparative models [355]. Apart from comparing model performance, the comparison of model architecture, data processing pipelines, and evaluation methods are also vital to propagating innovation. To establish a valid comparison within a single study, some scholars have used ablation studies to evaluate the individual components' importance in the overall modeling process [294]. In ablation studies, individual model components, feature inputs, or processing steps are removed to assess their importance. These studies, along with in-model comparisons, should be encouraged for all future studies to assess the effectiveness of the wide variety of model structures, techniques, and pipelines.

### 9.2. Training Protocols

As a data-driven approach, the practical application of deep learning to AD classification or prediction typically consists of a model or framework that acts as the basis of training to achieve the objective of individual algorithmic implementations. For most prediction and classification studies, this implies that the training and evaluation protocol has limited dependence on core architecture and utility. The protocol followed by training and evaluation impacts the models' performance, quality, and potential generalizability. This section will first introduce typical training and evaluation protocols, in addition to those methods mentioned in Section 7.1, and then highlight the hazard of information leakage. Then we will discuss appropriate optimization methods and the use of comparative studies.

### 9.2.1. Training and Evaluation Protocols

Standard training protocols involve using a single type of hold-out or cross-validation. However, using a validation set pre-selected from the training set or performing cross-validation on the training set for hyperparameter optimization is common. The metrics of performance on the validation set or CV of the training set provide a basis for the optimization. The use of a validation set is more commonly applied due to lower computational costs. When cross-validation is performed on the training set, testing can be performed on either a model from the cross-validation process or by retraining a new model with the entire training set.

More complex training and evaluation protocols can also be applied with sufficient data samples and computational resources. These protocols include component-wise parameter optimization, where each component of a neural network framework is trained or optimized separately. Random seeding is an evaluation procedure commonly used in machine learning, where multiple tests are run with different initial seeds for the ran-

dom number generators. As an interdisciplinary field between computer science and medicine, another procedure for evaluating model performance is human evaluation by medical practitioners.

### 9.2.2. Information Leakage

A significant concern under-addressed in current machine learning research in the field of AD or other diseases is the problem of information leakage [89]. This refers to the leakage of information from the training or validation set to the test set, introducing bias that can skew or invalidate the testing results. Information leakage can be categorized into three main types: (1) lack of test set, (2) invalid split, and (3) leakage in design.

The lack of an independent test set means the study cannot evaluate overfitting and generalization. The test set can also be invalidated if this subset is involved in hyperparameter optimization instead of a separate validation set. In these scenarios, the metrics do not provide a valid approximation of the model's actual performance and generalizability. Reported performance in these studies is typically significantly overstated quantitatively and should be considered as training performance only.

A test set can also be biased due to an invalid splitting process. A cross-sectional study might use the same subject's data samples at different time points as independent samples with longitudinal datasets in multiple data sources. However, if the split is performed according to images instead of subjects, the anatomical features of individual subjects could introduce bias that overfits the model at the subject level. A relative performance difference of 8% was found by Bäckström et al. [357]. Similar scenarios related to the invalid splitting of training and testing sets can also occur in other stages of the training process, e.g., data augmentation of an entire dataset before sample-wise splitting.

Apart from an invalid test set, information leakage can also occur through other factors in the studies. These factors include flawed data sourcing, where the same individual in multiple data sources is treated as independent. This leakage is possible for data sources such as ADNI, where some individuals are enrolled in multiple rounds of data collection and separated into different cohorts. Similar problems can occur with transfer learning where the source and target domains contain an amount of overlap. Information leakage is not limited to between the test and training sets but can also occur with intermediate subsets such as the validation set. With the validation set as an intermediate measure of model performance between the training and independent test set, we expect to utilize the validation set and, therefore, leak information. However, extensive hyperparameter optimization on the validation set can cause more information to be passed from this set to the model, causing overfitting on the validation set. Validation overfitting can negatively impact testing set performance and the overall generalizability of the model.

### 9.2.3. Optimization Protocols

Optimization is an essential component of overall training protocols and is divided into two main parts: optimization of parameters in the training process, and optimization of hyperparameters in the training protocol. Optimizers have become an essential component of current neural networks as they dictate the trajectory and means of gradient descent. Today's main optimizers are standardized, such as the stochastic gradient descent with momentum (SGDM) and RMSProp. Recent developments in neural network optimizers such as Adam and Adadelta have reduced dependence on learning rates and are more adaptive. Some studies include custom modifications, while others rely on model-dependent machine learning optimizers such as Limited memory BFGS [198,234]. The other main optimization component is the choice of hyperparameters, which define the overall structure and training specifications. Current methods are mostly based on grid search and random search, where a definite or random selection of hyperparameters is combined to train a model and evaluate performance. Statistical methods such as Bayesian optimization have only limited success due to the large search space and high computational cost.

*9.3. Development Platforms*

The modern development platforms are mainly based on MATLAB, R, and Python. MATLAB is a proprietary computational platform for science and engineering; its Deep Learning Toolbox provides an optimized framework for the efficient development and deployment of neural networks. The MathWorks File Exchange provides a platform for open-source code sharing, but the core platform is inherently closed-source. The most popular programming language for deep learning implementation and research is Python. The large open-source community has provided researchers with deep-learning libraries such as Tensorflow [358], Caffe [359], Theano [360], and PyTorch [361].

Higher-level APIs for these packages, such as Keras for Tensorflow and Fast.ai for Pytorch, have also been developed to lower scripting requirements and difficulty for researchers outside the field of bioinformatics and computer science. The majority of AD-related open-source deep learning packages or scripts are in Python. Keras is also available as a package for R, a popular programming language and platform for statistical computing and graphics. The access, availability, and reduced application difficulty of these platforms promote research into Alzheimer's disease and related diseases from an interdisciplinary perspective.

## 10. Path to Interpretation of Deep Learning Models

A significant challenge in applying DL to AD research is the lack of interpretability inherent in these often over-parameterized and highly complex data-driven models. A large number of studies have attempted to improve interpretability from different perspectives. Basic interpretation can be achieved through simple methods, e.g., correlation analysis and clustering of neural network features or predictions. Lin, Wu, Wu and Wu [333] analyzed the correlation between prediction error and individual features to validate the relationship between APOE-e4 and brain aging. Ding, Sohn, Kawczynski, Trivedi, Harnish, Jenkins, Lituiev, Copeland, Aboian and Mari Aparici [175] performed t-distributed stochastic neighbor embedding (t-SNE) on neural network-generated features to validate the model's understanding of AD disease stages, and a similar analysis with additional principal component analysis was performed by Son, Oh, Oh, Kim, Lee, Roh and Kim [177].

These simple methods offer a primer to the various methods utilized in the surveyed studies to explain model predictions and improve interpretability. In machine learning, the approaches to interpretability can be categorized into post hoc and intrinsic. Post hoc interpretation methods refer to probing and manipulation after the model is trained, while the intrinsic approach attempts to build a level of interpretability directly into the model architecture. However, since neural networks are inherent "black boxes," most deep learning methods surveyed focus on the post hoc approach. In this section, we detail three branches of the post hoc approach: (1) data-based methods, (2) architecture-based methods, and (3) model-agnostic visualization methods.

How data are processed and inputted into deep neural networks can fundamentally impact interpretability. ROI-based methods can provide a level of basic interpretability, which can be further translated to ROI sensitivity and feature stability. These measures can be projected to functional regions [198]. Feature maps can also be directly projected to ROIs [234]. Similarly, patch-based methods have some basic interpretability, e.g., Liu, Cheng, Wang, Wang and Initiative [170] visualized network attention areas by finding critical local patches that significantly affect class prediction probability, i.e., a drop in performance if they are removed. Graphical data also provide benefits for interpretability. Li, Rong, Meng, Lu, Kwok and Cheng [286] used analytic graph measures such as PageRank to determine the importance of each vertex in the input graph data, while Ju, Hu and Li [234] utilized brain networks of fMRI imaging to isolate functional regions of importance. At the voxel level, Duc, Ryu, Qureshi, Choi, Lee and Lee [273] visualized independent components of individual component analysis results as saliency maps on MRI and utilized these maps as inputs for the classification of AD and regression of MMSE. Methods utilizing

the inherent types and properties of data are often hybrids of traditional machine learning and deep learning that attempt to gain advantages from both approaches.

The choice of neural network architectures can also impact interpretability. As a classic example, the decoder component of convolutional autoencoders often contains transposed convolutional layers, or deconvolution layers, which are often used to reconstruct the input from the encoded feature space. The deconvolution process generates reconstructed images and feature maps, which can be compared globally with the input image, or locally with the anatomical structures of the input image [238].

A prime example is the use of generative models, such as variational autoencoders or GAN, to generate representational reconstructions through averaging iterative generations and structural transformations [244]. Alternatively, neural network architectures, such as the fully convolutional networks, which replace fully connected layers with global average pooling and SoftMax, can be designed to generate probabilistic maps [275,276]. The hierarchical framework implemented by Lee, Choi, Kim, Suk and Initiative [321] also allows for abnormality detection at various levels of voxels, patches, and regions, which can be combined to form a unified regional abnormality map. Visualization and interpretation methods that are dependent on architecture are also fundamentally constrained by the rigidity of the architecture and may not be able to adapt to new data or modalities. The data-based and architecture-based methods can be considered partially ad hoc. However, most data-model frameworks do not intrinsically provide functionality for tracing the decision process from inputs to classification probabilities and, therefore, cannot be considered intrinsically interpretable.

Transformer technology is a relatively new and powerful technique. The main area of application for transformers is language-based tasks. In future Alzheimer's disease research, transformers can extract meaningful information from medical records, patient interviews, and research articles by applying natural language processing techniques. Their ability to capture long-range dependencies in sequential data makes them highly suitable for analyzing textual data related to Alzheimer's disease.

Furthermore, transformers offer the potential for multimodal fusion in Alzheimer's disease research. Integrating data from multiple modalities, including imaging data, genetic information, and clinical assessments, can provide a comprehensive understanding of the disease. Transformers can facilitate the fusion of diverse data sources, capturing complex interactions and uncovering hidden relationships between different data types. One notable advantage of transformers is their attention mechanism, which enhances explainability. By highlighting relevant regions in images or identifying important words in the text, attention weights provide insights into the model's predictions. This interpretability feature can be valuable for medical professionals in understanding and validating the results of transformer models.

Alternatively, model-agnostic techniques exist to visualize feature saliency. The probabilistic maps generated through FCN by Qiu, Joshi, Miller, Xue, Zhou, Karjadi, Chang, Joshi, Dwyer and Zhu [275] are examples of CAM's dependence on the model structure. Recent developments of Grad-CAM utilize gradient information, allowing visualization of feature maps of various layers throughout the neural network. Tang, Chuang, DeCarli, Jin, Beckett, Keiser and Dugger [267] utilized a guided Grad-CAM with feature occlusion to identify amyloid-β plaques on immunohistochemically-stained slides. Similar methods were also applied to whole-brain MRI and identified GM regions around the hippocampus and ventricles that were consistent with anatomical pathology [281,290,322]. Monitoring model output with perturbations in the input is another method to interpret neural network function. An example of this approach is the swap test proposed by Nigri et al. [362], where patches of the image of interest are replaced by patches from reference images of an alternative class. The hippocampal region showed the highest impact on model predictions for the swap test. Mean relevance maps can also be generated for each category to interpret disease stages and progression from the perspective of groups instead of individuals [355]. Regional saliency maps were also combined with hippocampal segmentation by Liu, Li,

Yan, Wang, Ma, Shen, Xu and Initiative [169], while attention maps can also be included in the neural network framework to enhance performance and localization results [363]. However, as with the methods mentioned above, providing quantitative assessments of these visualizations is difficult.

Data-based methods, architectural-based methods, and model-agnostic visualization techniques are all constrained by their fundamental limits, e.g., the information content of patch-based ensembles is limited by the patch dimensions. The generative models summarized in Section 6.2 emphasize new approaches that dedicate modeling to interpretation by changing the core aim to visual attribution [250] and designing neural networks that are inherently interpretable, e.g., invertible neural networks [176]. These approaches present the most novel aspects on the path to interpretation. A basic summary is presented in Figure 11.

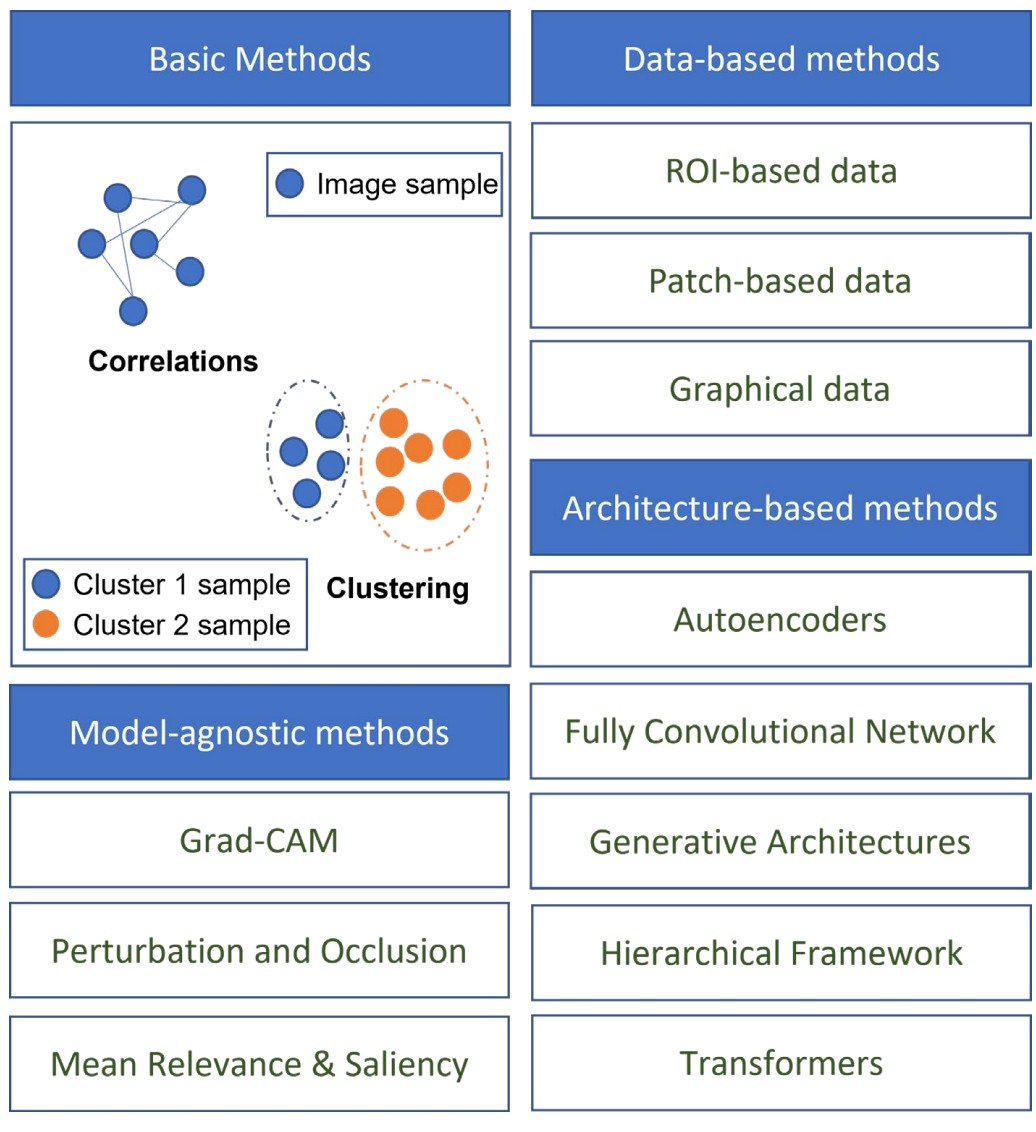

**Figure 11.** Overview of common interpretation methods found in surveyed literature.

## 11. Path to Generalization in the Real World

A major challenge in DL is the real-world generalization of models. Generalization is heavily affected by the data used to train the models. Most of the literature surveyed utilized data collected under strict acquisition protocols with specified modalities, types, and hardware and are often not representative of clinical settings. Preprocessing is commonly applied to eliminate some of these variabilities, but its variations and subjectivity can introduce uncertainty and different levels of quality and is, therefore, a double-edged sword. Mårtensson, Ferreira, Granberg, Cavallin, Oppedal, Padovani, Rektorova, Bonanni, Pardini and Kramberger [354] extensively assessed training and testing in different data domains.

While the proposed recurrent CNN showed consistency across different datasets, the recent study confirmed the expected degradation of performance in evaluating data collected through protocols that differ from those used for training data. As a solution, including a broader range of protocols in training, increased generalization performance in unseen data. Though this study is limited to a single CNN-based model and does not provide a definite conclusion, it provides valuable insight into the possible generalization challenges and the importance of data heterogeneity in countering them. It is established that generalization is heavily affected by the amount of data used in training and evaluation. Apart from collecting new data, methods to increase data quantity and heterogeneity during training include implementing lower-dimensional data (i.e., use of 2D slices of 3D scans), data augmentation, and the careful use of generative models. An alternative approach focuses on reducing the model's training data requirement, utilizing a train-test split of 50% or lower. These approaches are often semi-supervised and provide a larger testing set and a more accurate approximation of generalizability.

The theoretical forefront of this problem lies in estimating the 'generalization gap,' the difference between metrics derived from the independent test set and real-world scenarios. The approximate generalization bounds derived from the Hoeffding inequality [364] are based on a range of assumptions but provide a core insight into the relationship between the testing set and the approximate generalization gap—the bound is proportional to the inverted root of the sample size. Even though the amount of data required by these worst-case bounds is likely impossible to achieve in practical data collection, generalizability benefits from a larger sample size. A complete estimation is based on model complexity, usually measured through the Vapnik–Chervonenkis dimension. Alternative methods to derive generalization bounds have been explored, which include using the validation set [365], measurement of network smoothness [366], and comparison of generalization error between deep neural networks and humans [367]. An alternative approach to estimating generalization is to tackle the concept of label inhomogeneity due to misdiagnosis. Wu et al. [368] proposed using unsure data models to account for discordant MCI samples for which conversion is uncertain.

Apart from the technical and theoretical pathways to generalization, another key consideration is the practical generalization to clinical use, especially in mass screening. False positives produced by deep learning models in small-scale studies have been found to increase radiologist workload. In large-scale screening, the overdiagnosis caused by the number of false positives may significantly affect cost and efficiency [369,370]. Close monitoring of false-positive rates alongside generalization gap approximations should be a key aspect of evaluation in these scenarios.

## 12. Conclusions

In the past 13 years, many deep-learning studies have been conducted for AD and related diseases, producing various techniques, models, and protocols. We have provided a comprehensive summary of these major components that contribute to a deep learning study and a summary of the most recent advances, including recurrent neural networks, graph and geometric neural networks, as well as generative modeling.

These studies have shown promising results for a broad range of tasks, including image processing, disease categorical classification, and disease progression prediction. However, the wide variety of approaches shows a lack of consistency, and few studies provide standardized benchmarks for comparison. Most of these studies are research-oriented; few studies have conducted or simulated evaluations in clinical settings. These issues contribute to the challenges of interpretation and generalization of deep learning.

This review provides a glimpse into the possible solutions for interpretation, e.g., visualization techniques and inherently interpretable architectures. It also provides insights into potential pathways for generalization, e.g., data heterogeneity, data quantity, and generalization gap approximation. Apart from the key aspects of interpretation and generalization of neural networks, there are many other aspects of potential research, e.g., deep learning for polygenic studies [371] and the application of transformer-based foundational models. Combined with the continuously developing model architectures, these pathways will guide us toward more robust and clinically feasible deep-learning models for AD and related diseases.

**Author Contributions:** Q.Z.: Conceptualization, Software, Formal analysis, Writing—Original Draft, Visualization. J.W.: Software, Validation, Investigation, Writing—Original Draft, Visualization. X.Y.: Methodology, Formal analysis, Resources. S.W.: Methodology, Validation, Resources, Writing—Review and Editing, Supervision, Funding acquisition. Y.Z.: Conceptualization, Investigation, Writing—Review and Editing, Supervision, Project administration, Funding acquisition. All authors have read and agreed to the published version of the manuscript.

**Funding:** This paper is partially supported by MRC, UK (MC_PC_17171); Royal Society, UK (RP202G0230); BHF, UK (AA/18/3/34220); Hope Foundation for Cancer Research, UK (RM60G0680); GCRF, UK (P202PF11); Sino-UK Industrial Fund, UK (RP202G0289); LIAS, UK (P202ED10, P202RE969); Data Science Enhancement Fund, UK (P202RE237); Fight for Sight, UK (24NN201); Sino-UK Education Fund, UK (OP202006); BBSRC, UK (RM32G0178B8).

**Data Availability Statement:** Not applicable.

**Conflicts of Interest:** The authors declare no conflict of interest.

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
