# Peer review of "A Survey of Deep Learning for Alzheimer’s Disease"

_make, doi:10.3390/make5020035_

Round 1

Reviewer 1 Report

- Title must be clarified.

- Abstract and coclusions  should be revised

- Resolution of Figures can be enhanced.

English writting can be improved

Can be improved

Author Response

  1. Title must be clarified.

Answer:

We sincerely apologize for any confusion caused by the title. We have modified the title to “A Survey of Deep Learning for Alzheimer's Disease”. We hope the new title provides better clarity and is more representative of the content and scope of the paper.

  1. Abstract and conclusions should be revised

Answer:

We have revised both the abstract and conclusions, improving their clarity, coherence, and overall effectiveness in summarizing the key findings and implications of the research.

  1. Resolution of Figures can be enhanced.

Answer:

We understand the importance of clear and visually appealing figures, and thus we have worked diligently to enhance their resolution, making them more detailed and easier to interpret.

  1. English writing can be improved.

Answer:

We have dedicated substantial effort to improving the quality of English writing throughout the paper, resulting in higher clarity, coherence, and overall readability.

Thank you for your most valuable feedback!

Reviewer 2 Report

Authors has shown a compreshive review on this author. Great effort has been shown. Here are the comments below.

1. Figure 1, DATA pre-processing title is misleading. The content is not pre-processing technique. The position of table is poorly organized.

2.  “Improvement and a range of challenges” should be summarized in form of a figure with weights of challenges instead of a list only.

3. Survey protocol is unclear. For instance, what is recognized medical journal? Concentrated search needs more specific written and provide criteria of journal and conference selected such as impact factor>10. Key words for search should be provided. 360 papers were found and how there papers were filtered out in each stage with respect to the listed 6 points. It should be summarized in form of table or flowchart.

4. Sample figures of raw data and processed imaging modalities must be included in section 3.

 5. Summarize of performance of different machine learning method over different modalities must be included. Table 2,3, and 4 must be modified.

 6. The relation of Alzheimer’s disease studies and training and evaluation is not highlighted.

 7. Perspective of transformer technique application in future of Alzheimer’s disease study should be discussed in section 11.

In general, more figures should be included to support illustration.

Independent English editing should be included for such long narrative review.

Reviewer 3 Report

The paper explores the application of deep learning in the field of Alzheimer's disease and related disorders. It addresses the significant health issues posed by these diseases and highlights the growing interest in using deep learning techniques across various disciplines. The paper conducts a comprehensive survey of the literature on deep learning methods applied to Alzheimer's disease, mild cognitive impairment, and related conditions from 2010 to early 2023.

The paper provides a thorough review of different types of deep learning approaches, including unsupervised, supervised, and semi-supervised methods. It specifically focuses on recent advancements in the field, such as recurrent neural networks, graph-neural networks, and generative models. By covering these diverse techniques, the paper offers a valuable resource for researchers seeking to understand and utilize deep learning in the context of Alzheimer's disease.

In addition to discussing the various deep learning methods, the paper also provides a comprehensive summary of data sources, data processing techniques, training protocols, and evaluation methods employed in the reviewed studies. This comprehensive overview serves as a practical guide for future deep learning investigations in Alzheimer's disease research, facilitating the design and implementation of rigorous and effective studies.

The paper acknowledges the interpretational and generalization challenges associated with deep learning and highlights them as limitations of the approach. By addressing these challenges, the paper provides valuable insights and suggests potential pathways for future research endeavors in this domain.

Overall, the paper successfully communicates its objectives, scope, and key findings. It presents a well-structured survey of deep learning applications in Alzheimer's disease research, providing readers with a comprehensive understanding of the field's advancements and challenges. The paper serves as a valuable resource for researchers and practitioners interested in leveraging deep learning techniques to advance the understanding and management of Alzheimer's disease and related disorders.

Author Response

The paper explores the application of deep learning in the field of Alzheimer's disease and related disorders. It addresses the significant health issues posed by these diseases and highlights the growing interest in using deep learning techniques across various disciplines. The paper conducts a comprehensive survey of the literature on deep learning methods applied to Alzheimer's disease, mild cognitive impairment, and related conditions from 2010 to early 2023.

The paper provides a thorough review of different types of deep learning approaches, including unsupervised, supervised, and semi-supervised methods. It specifically focuses on recent advancements in the field, such as recurrent neural networks, graph-neural networks, and generative models. By covering these diverse techniques, the paper offers a valuable resource for researchers seeking to understand and utilize deep learning in the context of Alzheimer's disease.

In addition to discussing the various deep learning methods, the paper also provides a comprehensive summary of data sources, data processing techniques, training protocols, and evaluation methods employed in the reviewed studies. This comprehensive overview serves as a practical guide for future deep learning investigations in Alzheimer's disease research, facilitating the design and implementation of rigorous and effective studies.

The paper acknowledges the interpretational and generalization challenges associated with deep learning and highlights them as limitations of the approach. By addressing these challenges, the paper provides valuable insights and suggests potential pathways for future research endeavors in this domain.

Overall, the paper successfully communicates its objectives, scope, and key findings. It presents a well-structured survey of deep learning applications in Alzheimer's disease research, providing readers with a comprehensive understanding of the field's advancements and challenges. The paper serves as a valuable resource for researchers and practitioners interested in leveraging deep learning techniques to advance the understanding and management of Alzheimer's disease and related disorders.

Answer:

Thank you for your review. We appreciate your positive evaluation of the objectives, scope, and key findings of this paper. We believe that the paper contributes to the field by providing a thorough exploration of deep learning techniques in the context of Alzheimer's disease and related disorders. Your feedback further validates the paper's value as a resource for researchers and practitioners in the field. We sincerely appreciate your feedback. If you have any further comments or questions, please feel free to let us know.

Thank you for your most valuable feedback!

Round 2

Reviewer 2 Report

Thanks for the update.

Figure 1 has improved but it seems the position of the tables is less organized and appropriately aligned.

Survey protocol is improved. 360 papers were found and a number of papers filtered during each stage should be summarized in the flowchart and main text.

A full list of search keywords must be included and present in the flowchart and main text.

The recongized medical journals and conferences involved in the search process must be fully listed in the main text in section 1.6

It is fine.

Author Response

See attached PDF

Reviewer 3 Report

  1. The 3D patches in figures 4 and 9 seem to have been misunderstood as 3D CNNs.

  2. When evaluating the results of a multi-classification model, it is not reasonable to use sensitivity and specificity as the evaluation metrics.

  3. It is misleading to categorize deep learning as solely supervised or unsupervised. For instance, CNN is not strictly supervised or unsupervised; it is a neural network capable of extracting features from images by dividing, pooling, and stacking smaller areas of the image.

Author Response

The 3D patches in figures 4 and 9 seem to have been misunderstood as 3D CNNs.

Thank you for your comment. Additional captions have been added to Figure 9 to ensure distinction between the 3D patches in Figure 4 and 3D feature maps produced by convolutional operations in Figure 9.

When evaluating the results of a multi-classification model, it is not reasonable to use sensitivity and specificity as the evaluation metrics.

Thank you for your feedback. The metric of sensitivity and specificity have been removed from multi-class classification results.

It is misleading to categorize deep learning as solely supervised or unsupervised. For instance, CNN is not strictly supervised or unsupervised; it is a neural network capable of extracting features from images by dividing, pooling, and stacking smaller areas of the image.

Thank you for your feedback. I understand that by decomposing deep learning architectures many of its components may be individually supervised or unsupervised. However, in this survey, we are looking at the main neural network or framework used by each study and making the distinction on that level, where supervision is defined with the use of ground-truth labels. This distinction follows previous surveys in this interdisciplinary field.

Additional comments clarifying this have been added to the introduction of Section 6.

In deep learning, no architecture is strictly supervised or unsupervised if we decompose them into their base components, e.g., feature extraction and classification components of convolutional neural networks. In this survey, the distinction is made based on the relationship between the optimization target of the main neural network or framework and ground truth labels.

Thank you for your most valuable feedback!

Round 3

Reviewer 3 Report

Thanks for all your responses, I think these answers make sense.